# An immunostimulatory glycolipid that blocks SARS-CoV-2, RSV, and influenza infections in vivo

Moriya Tsuji [1,2,11] ✉, Manoj S. Nair [1,2,11], Kazuya Masuda [1,2,11], Candace Castagna[3,11], Zhenlu Chong[4], Tamarand L. Darling[4], Kuljeet Seehra[4], Youngmin Hwang[5], Ágata Lopes Ribeiro[6], Geovane Marques Ferreira[6], Laura Corredor[3], Jordana Grazziela Alves Coelho-dos-Reis[6], Yukiko Tsuji[1], Munemasa Mori[5], Adrianus C. M. Boon [4,7,8], Michael S. Diamond [4,7,8,9], Yaoxing Huang [1,2] ✉ & David D. Ho [1,2,10] ✉

Prophylactic vaccines for SARS-CoV-2 have lowered the incidence of severe COVID-19, but emergence of viral variants that are antigenically distinct from the vaccine strains are of concern and additional, broadly acting preventive approaches are desirable. Here, we report on a glycolipid termed 7DW8-5 that exploits the host innate immune system to enable rapid control of viral infections in vivo. This glycolipid binds to CD1d on antigen-presenting cells and thereby stimulates NKT cells to release a cascade of cytokines and chemokines. The intranasal administration of 7DW8-5 prior to virus exposure significantly blocked infection by three different authentic variants of SARS-CoV-2, as well as by respiratory syncytial virus and influenza virus, in mice or hamsters. We also found that this protective antiviral effect is both host-directed and mechanism-specific, requiring both the CD1d molecule and interferon-γ. A chemical compound like 7DW8-5 that is easy to administer and cheap to manufacture may be useful not only in slowing the spread of COVID-19 but also in responding to future pandemics long before vaccines or drugs are developed.

Although the COVID-19 pandemic has devastated humanity for the past three years, the scientific response to develop and deploy countermeasures against the causative agent, SARS-CoV-2, has been truly unprecedented. In record time, antiviral drugs[1,2] and monoclonal antibodies[3,4] were added to the therapeutic arsenal. Likewise, many prophylactic vaccines were developed to harness the power of the adaptive immune response, with several shown to be highly protective against symptomatic infection[5,6]. However, as SARS-CoV-2 continued to spread and viral variants continued to evolve, vaccine-breakthrough infections have become a common occurrence, particularly after the

[1]Aaron Diamond AIDS Research Center, Columbia University Irving Medical Center, New York, NY 10032, USA. [2]Division of Infectious Diseases, Department of Medicine, Columbia University Irving Medical Center, New York, NY 10032, USA. [3]Institute of Comparative Medicine, Columbia University Irving Medical Center, New York, NY 10032, USA. [4]Department of Medicine, Washington University School of Medicine, St. Louis, MO 63110, USA. [5]Columbia Center for Human Development, Pulmonary Allergy & Critical Care Medicine, Department of Medicine, Columbia University Irving Medical Center, New York, NY 10032, USA. [6]Basic and Applied Virology Laboratory, Department of Microbiology, Federal University of Minas Gerais, Belo Horizonte, Minas Gerais, Brazil. [7]Department of Molecular Microbiology, Washington University School of Medicine, St. Louis, MO 63110, USA. [8]Department of Pathology and Immunology, Washington University School of Medicine, St. Louis, MO 63110, USA. [9]The Andrew M. and Jane M. Bursky Center for Human Immunology and Immunotherapy Programs, Washington University School of Medicine, St. Louis, MO 63110, USA. [10]Department of Microbiology and Immunology, Columbia University Vagelos College of Physicians and Surgeons, New York, NY 10032, USA. [11]These authors contributed equally: Moriya Tsuji, Manoj S. Nair, Kazuya Masuda, Candace Castagna. ✉e-mail: mt3432@cumc.columbia.edu; yh3253@cumc.columbia.edu; dh2994@cumc.columbia.edu

emergence of the Omicron subvariants that are antigenically the most distinct from ancestral strains[7,8]. Novel prevention strategies could help to slow the transmission of this viral pathogen worldwide, including approaches that exploit the host innate immune system and enable rapid control of infection. For example, mice treated before, or soon after infection with a combination of inhaled Toll-like receptor (TLR) 2/6 and 9 agonists (Pam2-ODN) were shown to be broadly protective against microbial pathogens including respiratory viruses[9]. These agents induce activation of the antigen-presenting cells (APC) resulting in the downstream release of antiviral cytokines that mediate the clearance of the virus. Here, we evaluated a Natural killer T (NKT) cell-stimulatory agent for respiratory virus infections. NKT cells are a subset of lymphocytes possessing features of both natural killer (NK) cells and $\alpha\beta$ T cells[10,11]. These cells form a key element of the innate immune response and may play a role not only in cancer[12–14] and autoimmune diseases[15], but also in protection against infections[16–19]. Some NKT cells possess a semi-invariant T-cell receptor (iTCR) and are thus called invariant NKT cells (iNKT cells), which recognize certain glycolipids bound to CD1d molecules on antigen-presenting cells (e.g., dendritic cells or DCs)[20] and B cells[21], thereby triggering a cascade of cytokines and chemokines[10,11] (Fig. 1a). There have been a number of studies showing the importance of iNKT cells against viral infections, such as CMV, retroviral infection, RSV and influenza[22–26], as well as the role of NKT cells in the context of adaptive anti-viral immune response[27–30]. The first CD1d ligand identified was a glycolipid termed $\alpha$-galactosylceramide ($\alpha$-GalCer)[31]. Since then, there have been more than a dozen $\alpha$-GalCer analogues being described to date[14,32–43], all of which are able to stimulate iNKT cells in the context of CD1d molecules, resulting in exerting activities against various infections, cancers and auto-immune diseases primarily in a mouse model. From a focused library of synthetic glycolipids, we have discovered an $\alpha$-GalCer analog, 7DW8-5 (Fig. 1a), which also stimulates iNKT cells in the context of CD1d, but exhibits even more potent immunostimulatory activity in both mouse and human iNKT cells in vitro[44]. Upon stimulation, iNKT cells not only secrete Th1 cytokines that have known antiviral effects[24,25,45], but also activate populations of NK cells and CD8+ T cells[14,46,47]. Each of these activated cell populations could release multiple cytokines, including interferon-γ (IFN-γ) that has known protective effects against several viral infections[30,48,49].

In this work, we show that the CD1d-iNKT cell-dependent effect of 7DW8-5[44,50] prevents infection by SARS-CoV-2, respiratory syncytial virus (RSV), and influenza virus in animal models.

## Results

### 7DW8-5 blocks SARS-CoV-2 infection

We tested the hypothesis that the immunostimulatory effect of 7DW8-5 could inhibit SARS-CoV-2 infection in mice. First, 7DW8-5 (2 microgram (μg)) was administered intravenously (IV) or intranasally (IN) to groups of BALB/c mice two days before each animal was challenged IN with $5 \times 10^4$ plaque-forming units (PFU) of the mouse-adapted MA10 strain[51] of SARS-CoV-2. A third group of mice received saline and served as controls, while a fourth group was given 7DW8-5 (2 μg; IN) at 2 h post-virus challenge. When compared to controls, both IV and IN pre-exposure administrations of 7DW8-5 resulted in significant protection against MA10 infection as manifested by preservation of body weight and striking reductions of infectious virus in lung tissues (Fig. 1b). A follow-up study was performed to assess the timing of the antiviral effect exerted by 7DW8-5 (2 μg; IN). Strong protection against MA10 infection was observed when this glycolipid was given to mice either one or two days prior to virus exposure, whereas the protection was only partial when given three days prior (Fig. 1c). The post-exposure administration of 7DW8-5 showed a partial protective effect (Fig. 1c). A dose-exploration experiment then demonstrated robust protection of 7DW8-5 (IN; 2 days prior) at doses of 0.5 μg or 2 μg, whereas the protection was suboptimal at a dose of 0.1 μg (Fig. 1d).

We next assessed whether the repeated dosing of 7DW8-5 could lead to possible NKT-cell anergy and thus the loss of its antiviral efficacy. To this end, 7DW8-5 (0.2 μg; IN) was administered to a group of mice every other day for 10 days (for a total of five doses) before MA10 challenge. Serving as a comparator, another group of mice received the same glycolipid (0.2 μg; IN) only once at two days prior to virus exposure. The observed protective effects were equivalent for the two groups (Fig. 1e), indicating lack of anergy with repeated dosing of 7DW8-5 under these experimental conditions. It has been shown that an intraperitoneal or intravenous administration of a large dose (2-5 μg) of an iNKT-stimulatory glycolipid ($\alpha$-GalCer) induced iNKT cell anergy that causes unresponsiveness in mice[52,53]. Therefore, we also tested a larger dose (2 μg) of 7DW8-5 glycolipid intranasally to mice every other day for 10 days before MA10 challenge (Supplementary Fig. 1). Although the degree of the antiviral activity reduced, the repeated administration of a large dose of the glycolipid could still exert protective effects. This finding was corroborated by earlier study showing that intranasal but not intravenous delivery of $\alpha$-GalCer permits repeated stimulation of natural killer T cells in the lung[54]. Finally, to evaluate whether the antiviral effect occurred at the site of virus inoculation, mice were again given 7DW8-5 (2 μg; IN) two days before MA10 challenge, followed by harvesting of nasal turbinates and lungs three days later. Marked reductions of SARS-CoV-2 infection were observed in both sets of tissues from mice treated with 7DW8-5 (Fig. 1f).

Lung histopathologic analyses with hematoxylin and eosin (H & E) staining before and after MA10 challenge[51] revealed early multifocal damage with an accumulation of inflammatory cells, sloughed epithelial cells, and plasma proteins in the airway lumens (Fig. 1g). Treatment with 7DW8-5 prevented such lung tissue damage. Immunohistochemistry (IHC) staining[51,55] for SARS-CoV-2 nucleocapsid revealed an intense staining at 3-day post-infection in the lung tissue of a saline-treated and MA10-challenged mouse, whereas there was reduced staining and no visible staining in the lung tissues of mice treated with a single dosing or repeat dosing of 7DW8-5, respectively (Fig. 1g). In fact, when the number of infected spots were counted, both single and multiple dosing of 7DW8-5 significantly reduced the number of spots (Supplementary Fig. 2), consistent with the observed reduction in lung viral titers (Fig. 1b–e).

### Breadth of protection

We also assessed the antiviral effect of 7DW8-5 against other SARS-CoV-2 strains, starting with the Omicron subvariants BA.1 and BA.5, which can infect wild-type mice without prior adaptation. However, this infection model typically resulted in no weight loss in the infected animals as well as lower levels of virus replication[56]. Compared to controls, administration of 7DW8-5 (2 μg; IN) 2 days before the virus challenge ($1.5 \times 10^5$ PFU) led to significant weight gain and complete or robust inhibition of virus replication in lung tissues (Fig. 2a, b). A similar experiment was carried out using the Delta variant, which cannot infect wild-type mice but can infect K18-human ACE2-transgenic mice as well as hamsters[57]. Once more, administration of 7DW8-5 (2 μg; IN; 2 days prior) significantly lowered the viral load in both lung and nasal tissues of the virus-challenged ($10^3$ PFU) K18 human-ACE2 transgenic mice (Fig. 2c). Similarly, 7DW8-5 (100 μg/kg; IN) given 2 days before Delta variant challenge ($10^5$ PFU) significantly decreased the viral load in lung and nasal tissues of treated hamsters (Fig. 2d). In both experiments, the protection against the Delta variant was less robust than previously observed for MA10 or Omicron subvariants in wild-type mice, perhaps due to differences in virus susceptibility of the animal models. For example, the hACE2 expression pattern is known to be different in K18-hACE2 Tg mice[58], likely making them more permissive to infection.

Given that 7DW8-5 had no direct antiviral activity against SARS-CoV-2 in vitro (Supplementary Fig. 3) and was likely stimulating a part

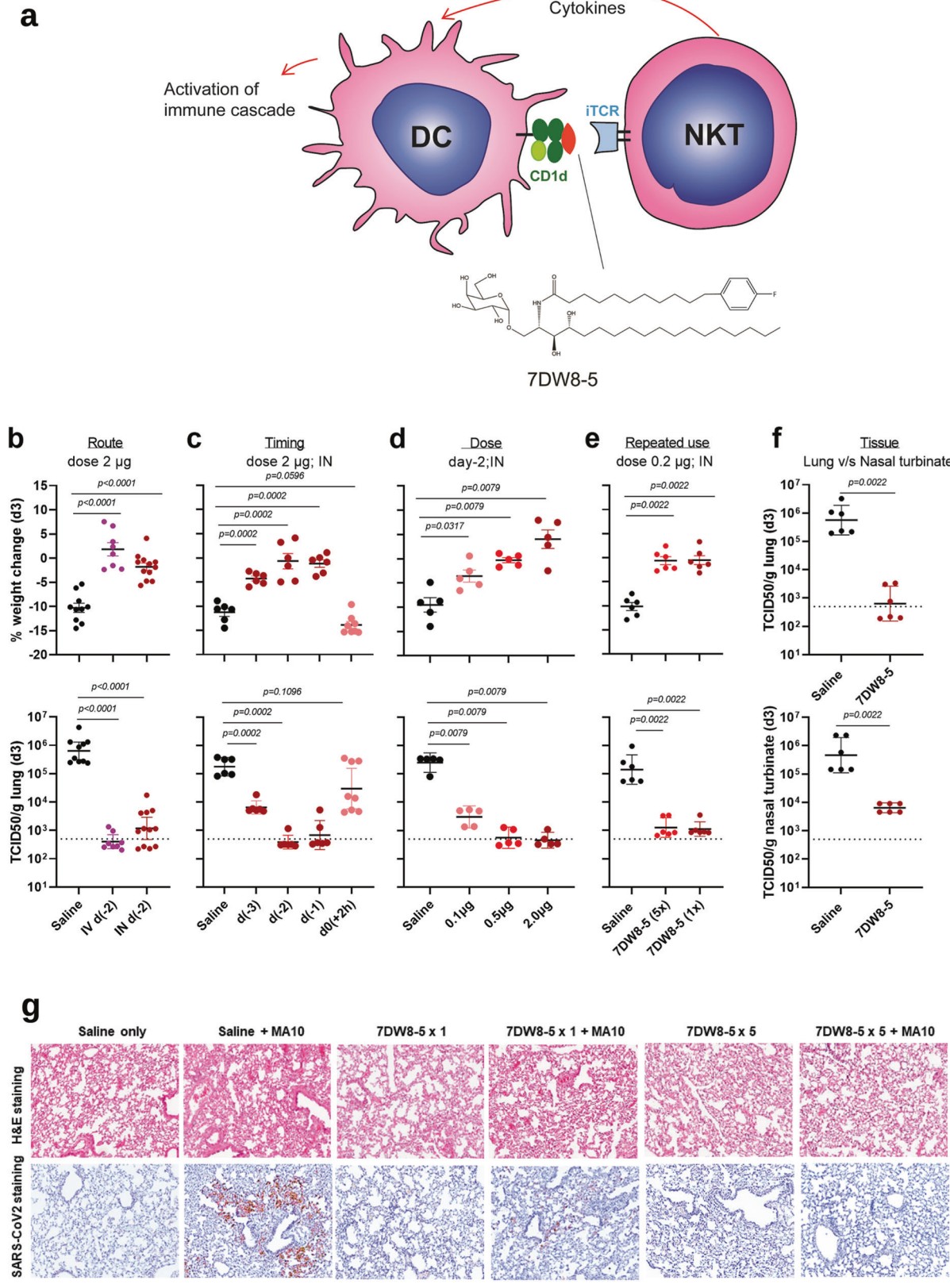

of the innate immune system, we then explored whether its protection could be extended to other clinically important viral pathogens: RSV and influenza virus. For the former, an RSV-A2 strain known to infect BALB/c mice was used[59]. As shown in Fig. 2e, administration of 7DW8-5 (2 µg; IN; 2 days prior) significantly prevented weight loss and inhibited virus infection in lung tissues of treated mice. For the influenza studies,

strain A/H1N1/PR8 known to infect C57BL/6 mice was used[60]. Again, administration of 7DW8-5 (2 µg; IN; 2 days prior) significantly prevented body weight loss and reduced the amount of infectious virus in lung tissues of treated mice (Fig. 2f). In a longer-term experiment using a lethal dose ($5 \times 10^3$ PFU) of influenza A/H1N1/PR8, all control mice died by Day 10 after virus exposure as expected, whereas 8 of 10 mice

**Fig. 1 | 7DW8-5 blocks SARS-CoV-2 MA10 (50,000 PFU) infection in BALB/c mice. a** Schematic showing the interaction between *i*NKT cells that carry iTCR and dendritic cells (DC) bearing CD1d that is bound to the glycolipid 7DW8-5 created using BioRender. Solid protection against infection, as manifested by maintenance of body weight and reduction of viral load (as determined by end-point dilution cultures) in lung tissues on Day 3 after virus challenge (d3) IN (intranasal), conferred by 7DW8-5 administered at (**b**) 2 µg IV (intravenous) or IN 2 days before challenge [d(-2)]. **c** Protection against infection conferred by 7DW8-5 administered at 2 µg IN 1 or 2 days before virus challenge [d(-1) or d(-2)] and less robustly if given 3 days before challenge [d(-3)], but not when given 2 h post-challenge [h(+2)]. (**d**) 0.5 or 2 µg IN but less so at 0.1 µg. **e** Repeat dosing of 7DW8-5 (0.2 µg; IN) every other day for 10 days (5x) conferred protection comparable to a single dose 2 days before

viral challenge (1x). **f** Protection against infection conferred by 7DW8-5 (2 µg; IN; 2 days before challenge) detected in both lungs and nasal turbinates. The dotted lines in the TCID50/g graphs indicate the limit of quantitation of the viral load. Non-parametric statistical analysis was done for experiments shown in (**b**) through (**f**), in Prism v 9.3 using two-tailed Mann–Whitney *U* test, and the results (including statistics) represent one of two independent biological experiments. Mean (black line) ± SEM is represented for each of the above graphs (**b**–**f**). **g** Representative 100 X images of lungs from 7DW8-5- and saline-treated mice before and after SARS-CoV-2 MA10 infection. H&E shown in the top panels. Bottom panels show immunohistochemistry (IHC) labeling against SARS-CoV-2 nucleocapsid, counterstained with hematoxylin.

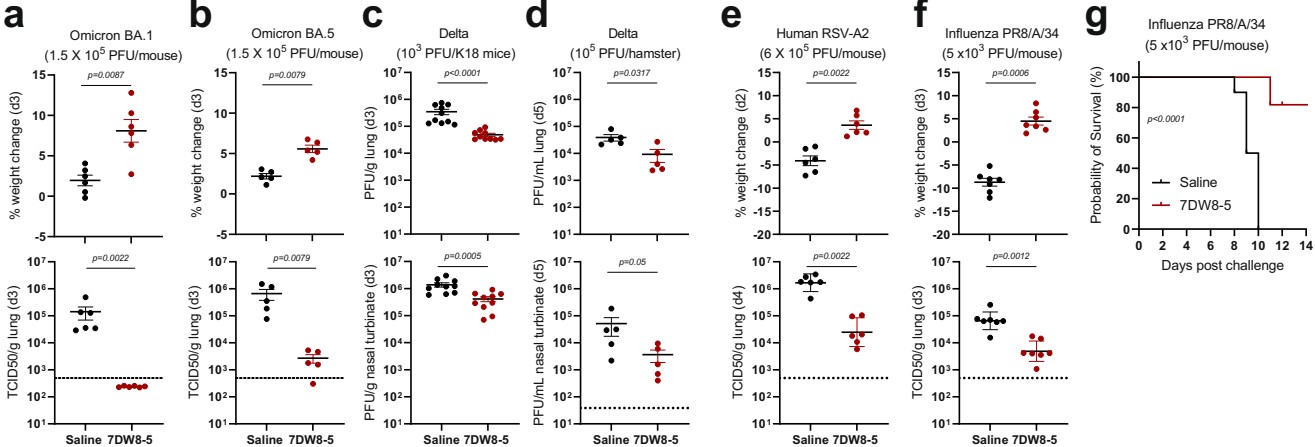

**Fig. 2 | 7DW8-5 has a broad antiviral activity against three different RNA viruses.** Breadth of protection conferred by 7DW8-5 (IN; 2 days before) extends to SARS-CoV-2 Omicron BA.1 and BA.5 variants in BALB/c mice (**a**, **b**) and Delta variant in K18 human-ACE2 transgenic C57BL/6 mice (**c**) or Syrian hamsters (**d**), as well as to RSV A2 strain in BALB/c mice (**e**) and influenza A1/H1N1/PR8 strain in C57BL/6 mice (**f**, **g**). The dose of challenge virus used is indicated at the top of each figure panel, and the number of animals used is reflected by the number of data points. The dose of 7DW8-5 given to mice was 2 µg each, whereas the dose for hamsters was 100 µg/kg. In (**a**, **b**), body weight was measured prior to and on day 3 post-viral challenge, and the infectious viral load was determined 3 days after viral challenge, as was done for (**c**) for both lung and nasal tissues. In (**d**), the viral load was determined 5 days after challenge. In (**e**), body weight was measured prior to and on day 2 post-

viral challenge, and the lung viral load was determined 4 days after the challenge. In (**f**), body weight was measured prior to and on day 3 post-challenge, and the lung viral load was determined 3 days after challenge. In (**g**), a Kaplan–Meier survival plot is shown following a lethal influenza virus challenge. Non-parametric statistical analysis was done for all the experiments shown in (**a**) through (**f**), in Prism v 9.3, using two-tailed Mann–Whitney *U* test, and the results (including statistics) represent one of two independent biological experiments. Mean (black line) ± SEM is represented for each of the above graphs (**a**–**f**). The dotted lines in the TCID50/g graphs and PFU/mL graphs indicate the limit of quantitation of the viral load. A log-rank Cox–Mantel test was performed in (**g**) to compare the survival curves of the two groups with a hazard ratio of 10.63 between them.

given 7DW8-5 survived (Fig. 2g). Taken together, the findings in Fig. 2 suggest that this glycolipid has substantial protective activity against divergent respiratory viruses, perhaps due to stimulation of a specific immune pathway, followed by a reciprocal stimulation of *i*NKT cell and CD1d-expressing DC activation.

**Cytokine induction**

We subsequently determined the level of gene expression of cytokines/chemokines in nasal turbinate tissue of BALB/c mice one day after administration of 7DW8-5 (2 µg; IN) or saline. We averaged three independent experiments with each comprising of 3 BALB/c mice. IFN-γ was the most upregulated cytokine in the nasal turbinate upon 7DW8-5 treatment (Fig. 3a). We also characterized the cytokine/chemokine profile in bronchoalveolar lavage (BAL), homogenates of nasal turbinates or lungs, and sera in BALB/c mice (*N* = 6) one day after administration of 7DW8-5 (2 µg; IN) or saline. Marked upregulation of multiple cytokines/chemokines was detected in all compartments examined from the treated animals, including IFN-γ, IL-5, IL-6, IP-10, TNF-α, MCP1, and MIG (Fig. 3b). IFN-γ, in particular, is a known antiviral cytokine[61]. Since IFN-λ has also been reported to have activity against SARS-CoV-2 in mice[62], we also measured the levels of this cytokine in the supernatants of nasal turbinate and lung homogenates, as well as in

sera, from mice treated with 7DW8-5 (2 µg; IN; 1 day prior) or saline. No evidence of IFN-λ induction was noted (Supplementary Fig. 4). A single-cell, RNA-sequence analysis of lung mononuclear cells from 7DW8-5-treated mice was conducted according to a specific experimental protocol (Supplementary Fig. 5a, b), and the expression levels of representative marker genes were documented within 18 distinct immune cell populations (Supplementary Fig. 5c). The most salient observation is the apparent shift toward IFN-γ production seen in proliferating NK cells and T cells, as well as in *i*NKT cells and γδ T cells, after exposure to 7DW8-5 (Fig. 3c). These findings implicate IFN-γ as possibly having a central role in mediating the antiviral activity of our glycolipid. Additional analysis of the *i*NKT cells identified in each UMAP of 7DW8-5 and saline (Fig. 3c) further categorized them into *i*NKT subsets including *i*NKT1, *i*NKT2, *i*NKT17 and *i*NKT10, based on the expression of the signature gene markers, as previously described[63]. A majority of *i*NKT cell subset, which had expanded among lung MNCs 3 days after intranasal administration (2 µg) of 7DW8-5, was *i*NKT1 (Supplementary Fig. 6a). Therefore, an *i*NKT1 subset, which secretes IFN-γ, appears to predominantly contribute to viral clearance. From our UMAP data (Fig. 3c), we have also categorized DC subsets and determined that 30% of mature DCs (mDCs) was found to express IL-15 among lung MNCs 3 days after intranasal administration (2 µg) of

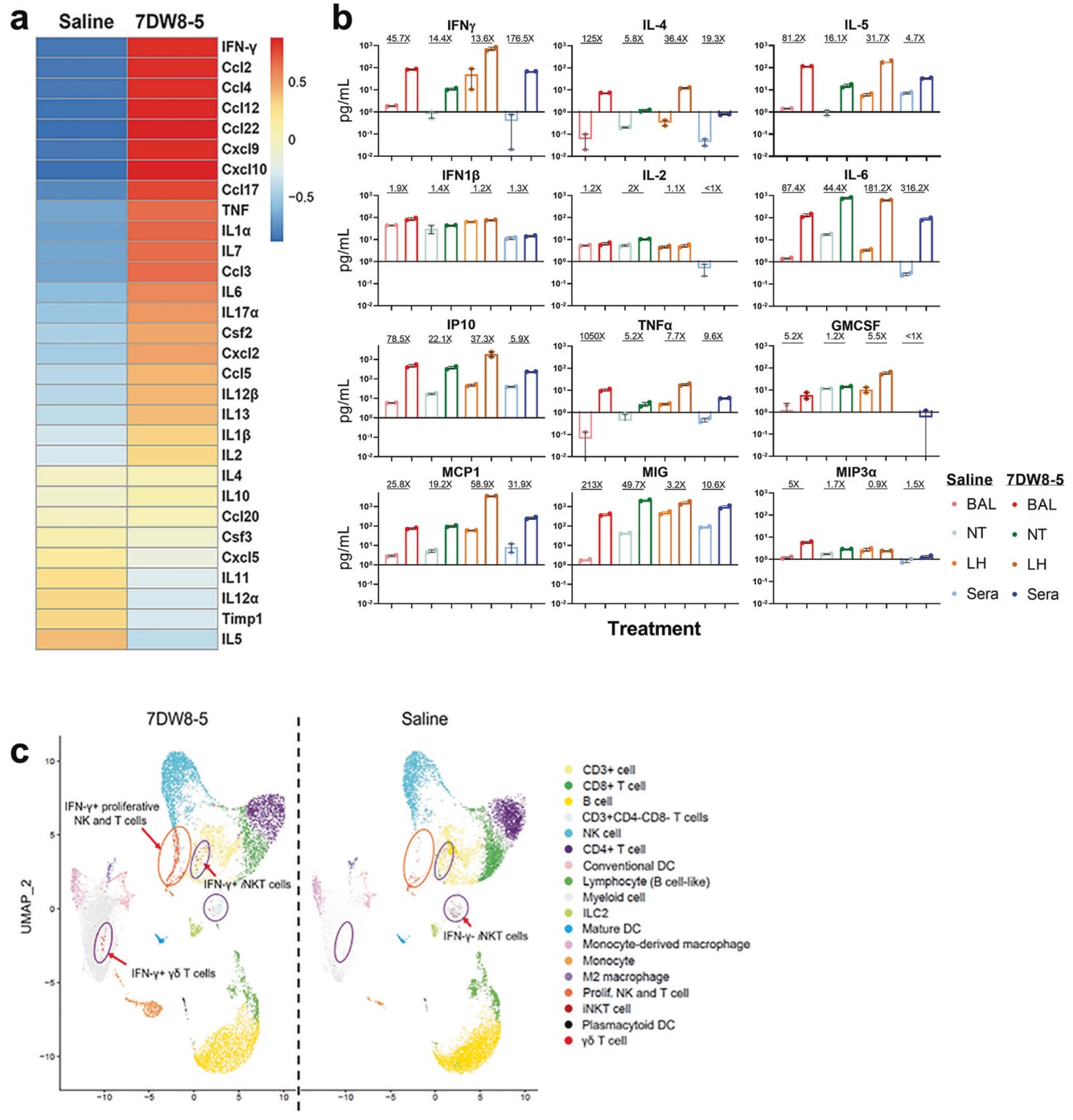

**Fig. 3 | Cytokine induction in BALB/c mice given 7DW8-5 (2 µg; IN). a** Heatmap showing the differential expression of the genes (as indicated) in nasal turbinates (NT), which is an average of three independent experiments with each consisting of a group of 3 BALB/c mice 24 h after 7DW8-5 (2 µg) or saline administration by IN. The scale bar shows the average Z-score. **b** Induction of cytokines/chemokines in bronchoalveolar lavage (BAL), supernatants of homogenates of lung (LH) and nasal turbinates (NT), and sera 24 h after 7DW8-5 administration to mice (*N* = 6), as assessed using the Mouse Cytokine Array/Chemokine Array 31-Plex (MD31) platform (Eve Technologies). Cytokine concentrations are plotted (in circles) and averages shown (using bar graphs), and the fold-change seen with 7DW8-5 treatment is expressed at the top. **c** Shift toward IFN-γ-producing cells as presented by Uniform Manifold Approximation and Projection (UMAP) of total 27,532 single CD45+ cells from the lungs from 3 BALB/c mice 24 h after treating with 7DW8-5 or saline (also see Supplementary Fig. 5a). The UMAP was split into 15,397 single cells (7DW8-5, left) and 12,135 single cells (saline, right). Each cluster (as indicated) was identified by unsupervised clustering and colored separately. *i*NKT cells were identified based on the expression of the semi-invariant TCR Vα14-Jα18.

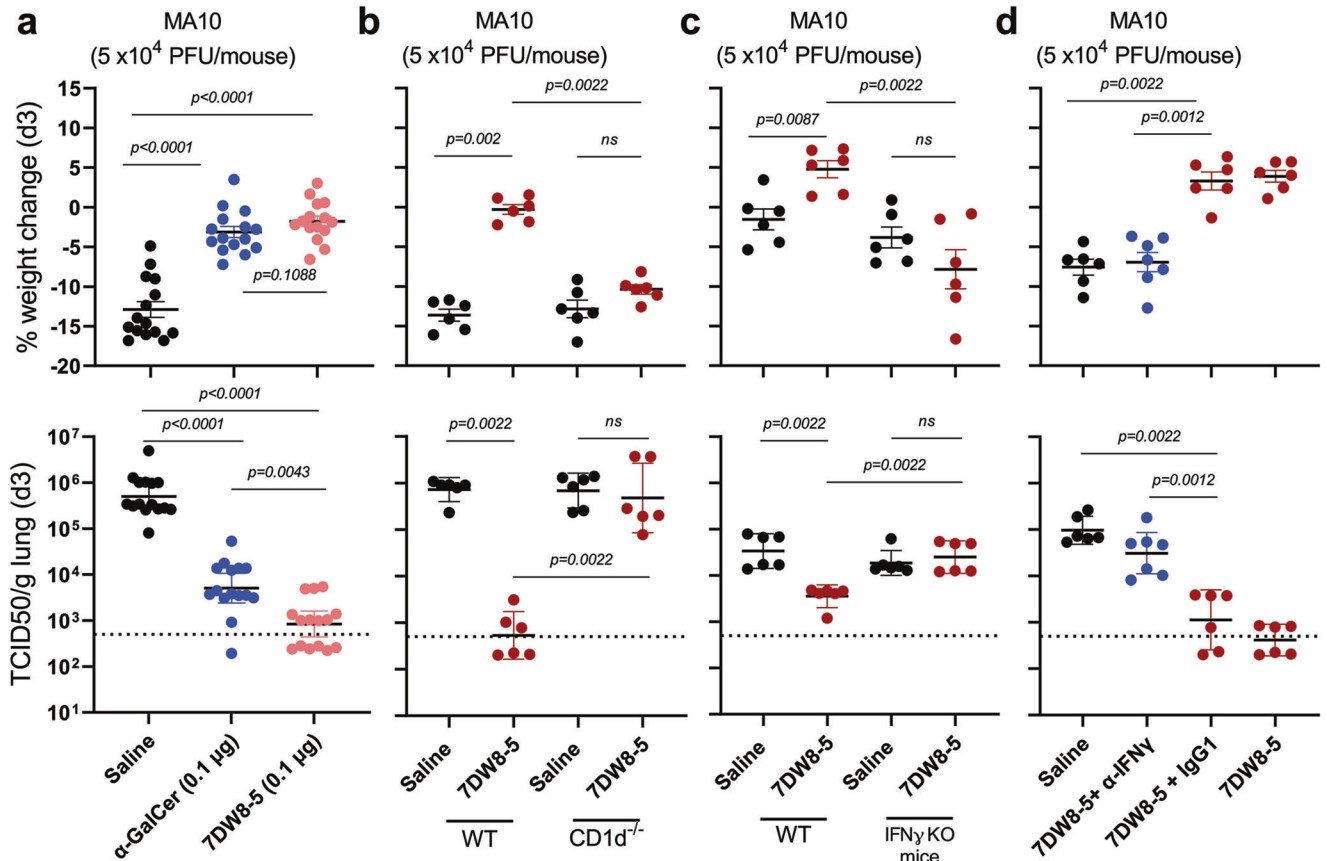

**Fig. 4 | Mechanism of Protection of 7DW8-5 against SARS-CoV-2 MA10. a** 7DW8-5 (0.1 μg; IN) and α-GalCer (0.1 μg; IN) to a lesser degree exert protective effect against SARS-CoV-2 in BALB/c mice when given 2 days before intranasal challenge with 50,000 PFU of the MA10 strain. Protection was based on body weight changes and infectious viral load in lung tissues on day 3 after virus challenge. The protective effect of 7DW8-5 is abolished in CD1d-KO BALB/c mice (**b**) or IFN-γ-KO C57BL/6 mice (**c**). **d** The protective effect of 7DW8-5 observed in wild-type mice is lost upon pre-treatment with a blocking anti-mouse-IFN-γ monoclonal antibody

(XMG 1.2 at 0.5 mg) but preserved upon intraperitoneal pre-treatment with a mouse IgG1 isotype-control monoclonal antibody (0.5 mg). 2 μg of 7DW8-5 was dosed in each animal used in (**b–d**). Non-parametric statistical analysis was done for all the experiments shown in (**a**) through (**d**), in Prism v 9.3 using two-tailed Mann–Whitney $U$ test, and the results (including statistics) represent one of two independent biological experiments. Mean (black line) ± SEM is represented for each of the above graphs. The dotted lines in the TCID50/g graphs indicate the limit of quantitation of the viral load.

7DW8-5 Supplementary Fig. 6b), whereas mDCs constitutively express IL-12 (data not shown). Plasmacytoid DCs (pDCs) and conventional DCs (cDCs) express minimal amount of IL-12 (data not shown) before and after 7DW8-5 treatment.

### Mechanism of protection

The final portion of this study addressed the mechanism of protection for 7DW8-5. First, the role of CD1d-mediated activation was examined by comparing 7DW8-5 to its active parental compound, α-GalCer, and to 1:18 caproylamine PE, which is known to bind to CD1d without stimulating *i*NKT cells due to the lack of a sugar moiety (Supplementary Fig. 7)[44,64]. Both 7DW8-5 and α-GalCer administration (0.1 μg; IN; 2 days prior) resulted in protection. However, there is a significant difference in lung viral load between the mice receiving α-GalCer versus 7DW8-5 upon MA10 challenge (Fig. 4a). The administration of 1:18 caproylamine PE at a much higher dose (2 μg; IN) failed to protect. This finding led to a subsequent experiment using CD1d knock-out (KO) mice. Indeed, the protective effect of 7DW8-5 observed in wild-type mice against SARS-CoV-2 MA10 challenge was completely abolished in CD1d-KO mice (Fig. 4b), demonstrating the indispensable role of CD1d activation in the antiviral activity of 7DW8-5.

Next, we determined the role IFN-γ played in contributing to the biological activity of 7DW8-5 by conducting protection studies in mice deficient in IFN-γ. Once again, the antiviral activity of 7DW8-5 observed in wild-type mice against SARS-CoV-2 MA10 challenge was totally lost

in IFN-γ-KO mice (Fig. 4c). Furthermore, the anti-SARS-CoV-2 effect of 7DW8-5 in wild-type mice was largely eliminated by the intraperitoneal injection of a blocking anti-mouse-IFN-γ monoclonal antibody prior to the glycolipid treatment (Fig. 4d). These latter results establish that IFN-γ is another essential mediator of the activity of 7DW8-5, along with CD1d.

The next imminent question relates to the translatability of our findings from small animal models to humans. CD1d molecule has been known to be highly conserved between humans and mice[20]. In fact, 7DW8-5 was able to noticeably stimulate primary human CD161[high]TCRVα24[+] NKT cells to secrete IFN-γ in vitro (Supplementary Fig. 8a–c). Most importantly, we found that the supernatants collected from human *i*NKT cell lines stimulated with CD1d-transfected Hela cells in the presence of 7DW8-5, but not PE, were able to significantly inhibit ic-SARS-CoV-2-mNeonGreen[65] infection in Huh7 cells in vitro[66], and that this inhibition was completely abolished when neutralizing anti-human IFN-γ Ab was added to the culture (Supplementary Fig. 8d). All these results indicate a significant role that IFN-γ plays for mediating anti-SARS-CoV-2 activity by 7DW8-5 both in mice and humans.

### Discussion

We have found that intranasal administration of 7DW8-5, an immunostimulatory glycolipid, worked well as pre-exposure prophylaxis in blocking SARS-CoV-2 infection in wild-type mice, as shown by maintenance of body weight and >2 log reduction of virus replication in the

lung (Fig. 1b–f), as well as by the marked reduction of viral nucleocapsid protein in the lung tissue (Fig. 1g). The viral load in the nasal turbinates was also reduced by ~50-fold (Fig. 1f), which is more than the blockade seen with monoclonal antibodies in this tissue compartment[67]. However, post-exposure administration of 7DW8-5 was ineffective (Fig. 1c), suggesting that this glycolipid will not be useful as a therapeutic, perhaps because its immunostimulatory effects need a sufficient head start in order to slow a fast-replicating virus. We also showed that the protection conferred by 7DW8-5 extended to three authentic SARS-CoV-2 variants, including Omicron subvariants BA.1 and BA.5 in wild-type mice (Fig. 2a, b) and the Delta variant in K18 human-ACE2 transgenic mice and hamsters (Fig. 2c & 2d). Moreover, comparable antiviral activity was observed in mice following challenges with RSV (Fig. 2e) or influenza virus (Fig. 2f, g). Overall, these findings demonstrate the breadth of protection of 7DW8-5 versus three families of respiratory viruses (coronaviruses, paramyxoviruses, and orthomyxoviruses), each of which is not only clinically important but also of pandemic potential.

The parental glycolipid, α-GalCer, is a potent activator of *i*NKT cells, inducing the production of large amounts of IFN-γ, which helps activate both CD8+ T cells and antigen-presenting cells (APCs), such as DCs, macrophages, and B cells[46]. We have previously shown that the action of 7DW8-5 is also dependent on CD1d and *i*NKT cells[44]. It is noteworthy that when a small dose (0.1 μg) of 7DW8-5 or α-GalCer was given to mice, followed by MA10 challenge, 7DW8-5 exerted a significantly more potent antiviral effect, resulting in a more reduced lung virus load (Fig. 4a). While many cytokines/chemokines were induced by this glycolipid (Fig. 3a, b), its antiviral effect was entirely abolished in CD1d-KO mice (Fig. 4b), which was anticipated since 7DW8-5 was selected for optimal binding to and triggering through both human and mouse CD1d (Fig. 1a)[44]. Similarly, another essential component of the protective effect was shown to be IFN-γ, a molecule in the innate immune system known to possess broad antimicrobial activities[61]. Indeed, 7DW8-5 showed no protection at all in IFN-γ-KO mice (Fig. 4c), and its protective effect in wild-type mice was largely lost when pre-treatment with a blocking anti-IFN-γ monoclonal antibody was given (Fig. 4d). Therefore, both CD1d and IFN-γ are necessary for the activity of 7DW8-5 in vivo; however, it remains unclear if both are sufficient because there may be important mediators downstream of IFN-γ.

It is our belief that 7DW8-5 holds promise as an agent for pre-exposure prophylaxis to prevent infections by SARS-CoV-2, RSV, influenza virus, and possibly other clinically important respiratory viruses. Its potential utility to future viral pandemics has not escaped us either. It is a chemical compound that is thermostable and thus simple to ship and store, cheap to manufacture, and easy to administer intranasally. Given our collective experiences with the COVID-19 pandemic, it is not hard for anyone to conjure up numerous everyday situations when high-risk exposures could be anticipated and when an effective preventive measure could be applied, if available. But would the results from rodents reported here be translatable to humans? A definitive answer could only come from the conduct of clinical studies, but there is reasonable expectation that 7DW8-5 could be similarly effective in people. First, we selected this glycolipid from a library of analogues based on its potent stimulatory effect on human *i*NKT cells in vitro[44]. 7DW8-5 displayed 80-times higher binding affinity to human CD1d than α-GalCer and exerted a 140-fold higher dose sparing effect against human *i*NKT cells over α-GalCer. Second, *i*NKT cells constitute up to 1% of peripheral blood mononuclear cells in both humans and mice[68,69], although the variability is greater among humans[68]. Lastly, we have shown in this study that the supernatant of human *i*NKT cells activated by 7DW8-5 can display anti-viral activity in vitro and that the activity is inhibited by anti-human IFN-γ antibody (Supplementary Fig. 8).

Many challenges will need to be overcome before 7DW8-5 could be considered a candidate for clinical development. Foremost among

them is its safety and tolerability, because the induction of cytokines, such as TNF-α and IL-6 (Fig. 3a, b), could result in an exaggerated inflammatory response. Formal safety/toxicity studies in two or more animal species will therefore be required. However, several observations address this concern. Prior clinical trials[70–73] using the parental glycolipid, α-GalCer, when administered (6 IV doses of 0.12 mg/kg) to cancer patients showed no evidence of toxicity[70]. In a vaccine adjuvant study in rhesus macaques, up to 100 μg of 7DW8-5 intramuscularly did not lead to any adverse reactions[74], similar to the lack of toxicity noted in mice and hamsters in the current study. Nevertheless, additional studies on 7DW8-5 will also be necessary to narrow the duration of its prophylactic effect, to determine more precisely its optimal dose, and to assess for evidence of anergy with its repeated usage over a protracted period.

Current COVID-19 vaccines have mitigated the impact of the pandemic by decreasing symptomatic infections, hospitalizations, and deaths. The excessive fear that was once rampant has largely dissipated due to the development of these vaccines as well as effective treatments. However, as SARS-CoV-2 has continued to evolve antigenically[75], breakthrough infections have become common[76], resulting, at a minimum, in major disruptions to our daily lives. Additional prevention modalities are therefore needed. If found to be safe, 7DW8-5 could be another tool in our fight against COVID-19 and other respiratory virus infections, which would also lead to a huge economical loss. The strength of the CD1d agonist would lie in the rapid responses to unknown emerging viral infections when vaccines and other virus-specific approaches are still under investigation. In this regard, there is also a possibility of an alternative use of 7DW8-5 glycolipid, as a vaccine adjuvant[74]. In conclusion, this immunostimulatory glycolipid could be applied practically using diverse strategies to respond to future outbreaks or pandemics caused by an emergent respiratory virus, preceding and during development of drugs and vaccines.

## Methods

### Data reporting
Power analysis based on guidelines by Institute for Laboratory Animal Research were used to predetermine sample size to estimate minimum number of animals required to detect significant effect of 7DW8-5 glycolipid, if one is detected. The experiments were not randomized, and the investigators were not blinded to allocation during experiments and outcome assessment.

### Ethics statement
Human peripheral blood mononuclear cells (PBMCs) from anonymous blood donors were obtained from leukopacks provided by the New York Blood Center (NYBC). The NYBC does not select donors on the basis of gender or race but ensures that all donors are above 18 years of age. Therefore, the work we performed did not require approval from the Institutional Review Board. All animal experiments were carried out in strict accordance with the Policy on Humane Care and Use of Laboratory Animals of the United States Public Health Service. The protocol was approved by the Institutional Animal Care and Use Committee (IACUC) at The Columbia University (Animal Welfare Assurance no. D16-00003) and the Washington University School of Medicine (Animal Welfare Assurance no. A3381–01). Virus inoculations were performed under anesthesia that was induced and maintained with ketamine hydrochloride and xylazine, and all efforts were made to minimize animal suffering. Mice and hamsters were euthanized with $CO_2$, with every effort made to minimize suffering.

### Animals
Female BALB/c mice with 10–15 weeks of age and female C57BL/6 mice with 14-15 weeks of age, as well as female BALB/c mice lacking CD1d1 and CD1d2 genes (strain: C.129S2-Cd1tm1Gru/J) and female C57BL/6

mice lacking IFN-γ (strain: B6.129S7-Ifngtm1Ts/J) were purchased from The Jackson Laboratory (Bar Harbor, ME) and maintained under specific pathogen-free conditions in the animal facility at Columbia University Irving Medical Center. Heterozygous 8-9 weeks old female K18-hACE C57BL/6J mice (strain: 2B6.Cg-Tg(K18-ACE2)2Prlmn/J) were obtained from The Jackson Laboratory and housed in a pathogen-free animal facility at Washington University School of Medicine. Male Syrian hamsters with 5–6 weeks of aged were purchased from Charles River Laboratories (Wilmington, MA) and housed in an enhance biosafety level 3 (BL3) facility at Washington University in St. Louis.

## Glycolipids
Glycolipid 7DW8-5 with chemical formula [(2S,3S,4R)-1-O-(α-D-galactopyranosyl)-N-(11-(4-fluorophenyl) undecanoyl)-2-amino-1,3,4-octadecanetriol)] was synthesized as previously described[44]. Glycolipid α-galactosyl ceramide (α-GalCer) with chemical formula [N-[(3S,4R)-3,4-dihydroxy-1-[(2S,3R,4S,5R,6R)-3,4,5-trihydroxy-6-(hydroxymethyl) oxan-2-yl]oxyoctadecan-2-yl]hexacosanamide] and control lipids with chemical formula 18:1 caproylamine PE [1,2-dioleoyl-sn-glycero-3-phosphoethanolamine-N-(hexanoylamine)] and 18:1 biotinyl PE [1,2-dioleoyl-sn-glycero-3-phosphoethanolamine-N-(biotinyl) sodium salt] were purchased from Avanti Polar Lipids.

## Cells and viruses
Vero-E6 (ATCC Cat #1586) and Vero-TMPRSS2-ACE2 cells (gift from Emory University) were cultured at 37 °C in Dulbecco's Modified Eagle medium (DMEM) supplemented with 10% fetal bovine serum (FBS), 10 mM HEPES pH 7.3, and 100 U/ml of penicillin–streptomycin. Huh7, a human hepatocarcinoma cell line was purchased from Japanese Collection of Research Bioresources (JCRB0403) and cultured in same conditions as Vero-E6 cells. The SARS-CoV-2 MA10, Delta variant (B.1.617.2) and Omicron variants, BA.1 and BA.5 viruses, were obtained from BEI (cat # NR-55329, NR-55691, NR-56475 and NR-58616). Omicron stock was propagated in Vero-TMPRSS2 cells as described[7] while other SARS-CoV-2 isolates were propagated in Vero E6 cells for this study as described[77]. The infectious clone ic-SARS-CoV-2, WA1/mNeonGreen was obtained from the World Reference Center for Emerging Viruses and Arboviruses (WRCEVA) at the University of Texas Medical Branch and propagated and titrated in Vero-E6 cells prior to use. All work with infectious SARS-CoV-2 was performed in approved BSL3 and A-BSL3 facilities at Columbia University Irving Medical Center or at Washington University using appropriate positive pressure air respirators and protective equipment. Madin-Darby Canine Kidney (MDCK) cells (Cat # CCL-34) and HEp-2 cells (Cat # CCL-23) were purchased from the ATCC. Influenza A virus (H1N1) PR/8/34 strain (Cat # VR-95) and human respiratory syncytial virus (RSV) A2 isolate (Cat # VR-26PQ), purchased from ATCC were propagated and titrated in MDCK and Hep-2 cells, respectively under BSL2 condition. Hep-2 cells purchased from ATCC (Cat # CCL-23) have been registered by the International Cell Line Authentication Committee as a cell line that is derived via HeLa contamination but has been validated for propagation of (RSV) A2 isolate in the laboratory prior to titration experiments.

## Nasal turbinate and lung homogenates
Nasal turbinate was harvested from mice as described[78]. Briefly, the skin was dissected and completely removed from the skull and nose, and the skull sectioned in the coronal plane. Then the remainder of the anterior skull was removed. After removing the posterior aspect of the skull base, the scissor was inserted into the posterior nasal cavity. The suture line was then incised bilaterally revealing the nasal septum. The scissor separated the two maxillas (laterally) and the ethmoid bone (medially) along the distinct suture line. Finally, the upper palate and anterior nasal tip were removed to complete the dissection. For the nasal turbinate from hamster, nasal turbinate was harvested by first

removing the skin along the side of the nose and cheeks. The jaw was then cut exposing the hamster's palate. A sagittal incision through the palate was made exposing the nasal turbinate which was removed via blunt forceps. The nasal turbinate was homogenized in 1 mL DMEM supplemented with 2% FBS, ʟ-glutamine and 1% HEPES. Homogenate was centrifuged at 1000$g$ for 5 min. Whole lung tissue was weighed after separation from bronchial tubes and other non-lung tissues, placed in BioMasher II tube (Diagnocine LLC) with 250 μL of DMEM media supplemented with 2% heat-inactivated FBS and manually homogenized by rotating the grinder back and forth until all the tissue was thoroughly homogenized. Afterwards, the tissue homogenates in the tubes were centrifuged at 1000$g$ at 4–10 °C for 10 min. For both nasal turbinate and lung homogenates, the supernatant was collected and stored at −80 °C for viral titer and viral load detection.

## Tissue viral load assays
Lung and nasal turbinate homogenates of 7DW8-5- or saline-treated and challenged mice and hamsters collected in the ABSL3 facility was used to set up the $TCID_{50}$ titration assay. Serial 2-fold dilution of the samples were added to a 96-well plate seeded with $2 \times 10^4$ Vero-E6 cells at 37 °C under 5% $CO_2$[77]. Three days later, the wells were visually scored for cytopathic effect (CPE, as observed by light microscopy) of the cells at each dilution, to determine the end point TCID. For the PFU assay, Vero-TMPRSS2-ACE2 cells were seeded at a density of $1.25 \times 10^5$ cells/well in flat-bottom 24-well tissue culture plates. The following day, media was replaced with 200 microliters of 10-fold serial dilutions of sample, diluted in DMEM + 2% FBS. One hour later, 1 mL of methylcellulose overlay was added. Plates were incubated for 72 h, then fixed with 4% paraformaldehyde (final concentration) in PBS for 1 h. Plates were stained with 0.05% (w/v) crystal violet in 20% methanol and washed twice with distilled, deionized water. Plaques were counted, and titers were calculated according to a previously described method[79].

For RSV titration, lung homogenates of saline or 7DW8-5 treated and challenged mice were serially diluted by 4-fold and overlaid on confluent layer of Hep-2 cells[59]. A virus control standard of the A2 isolate used to infect the mice was also serially diluted and run in parallel to compare the resulting CPE. Infected and control cells were incubated at 37 °C under 5% CO2 for 5 days. Viral CPE was observed under microscope for each dilution. Using the endpoint dilution method, the $TCID_{50}$ was calculated per treatment group.

For influenza PR8 virus titration, lung homogenates of saline- or 7DW8-5-treated mice were harvested on day-5 post challenge with 200 PFU of PR8/A/34 virus per mouse. Homogenates were serially diluted by 3-fold and overlaid on a confluent layer of Martin Delaney Canine kidney (MDCK) cells. The PR8/A/34 isolate also serially diluted and used as positive control. Cells were incubated at 33 °C/5% $CO_2$ and supernatants were collected at 72 h to determine viral neuraminidase activity from each well using a fluorescence assay[80,81] with the NA-Fluor Influenza Neuraminidase Assay kit (Applied Biosystems). Production of fluorogenic product 4-methylumbelliferone was read in a Spectramax i3X microplate reader using excitation wavelength of 360 nm and emission wavelength of 450 nm. Negative controls using uninfected cell supernatants serve as blank for the assay. Fluorescence data was collected using Softmax Pro 7.0.2 software. The endpoint for each sample was calculated as 3-fold or higher increase of the neuraminidase activity compared to the controls. TCID/g of lung was calculated for each sample and plotted using GraphPad Prism v9.3.

## Histology and immunohistochemistry
Histological and immunohistochemical assays were conducted as described[51,55]. Immediately after euthanasia, the left lung lobe was harvested and fixed by submersion in 10% phosphate-buffered formalin for overnight, followed by submersion in 70% ethanol for up to one week, which was enough to disinfect the viruses. Fixed tissue was

manually embedded in paraffin, and sections were prepared at 4 μm thickness. Tissue sections were stained with hematoxylin and eosin (Richard Allan Scientific, San Diego, CA). For immunohistochemistry, antigen retrieval was performed with antigen unmasking solution (pH 6.0 citrate buffer, H-3300, Vector Laboratories, Newark, CA), followed by peroxide incubation. The tissue sections were first incubated with primary antibody, SARS-CoV-2 nucleocapsid protein rabbit mAb (26369, Cell Signaling, Danvers, MA) for overnight, and then with secondary antibody, HRP-conjugated anti-rabbit IgG antibody (MP7401, Vector Laboratories). The signal was then amplified using a DAB staining kit (NC9567138, Fisher Scientific, Waltham, MA) and it was counterstained with hematoxylin. Finally, tissues were tiled scanned using an Aperio AT2 slide scanner (Leica Biosystems, Deer Park, IL), and images were captured with an Aperio ImageScope (Leica Biosystems, Deer Park, IL).

### RNA isolation and RT-qPCR
Total RNA was isolated from nasal turbinates with miRNeasy micro kit (Qiagen, 217084) according to the manufacturer's instructions. ERCC RNA Spike-In Mix kit (ThermoFisher Scientific, 4456740) was added to normalized total RNA prior to library preparation following the manufacturer's protocol. The RNA sequencing libraries were prepared using the NEBNext Ultra II RNA Library Prep Kit for Illumina according to the manufacturer's instructions (New England Biolabs, Ipswich, MA, USA). The samples were sequenced by the Illumina HiSeq instrument according to the manufacturer's instructions using a 2x150bp Paired End (PE) configuration. After the stringent QC of the raw data, sequence reads were trimmed to remove possible adapter sequences and nucleotides with poor quality using Trimmomatic v.0.36. The trimmed reads were mapped to the Mus musculus reference genome available on ENSEMBL using the STAR aligner v.2.5.2b. BAM files were generated as a result of this step. Unique gene hit counts were calculated by using feature Counts from the Subread package v.1.5.2. Only unique reads that fell within exon regions were counted. After extraction of gene hit counts, the gene hit counts table was used for downstream differential expression analysis. Using DESeq2, a comparison of gene expression between the groups of samples was performed.

### Treatment with anti-IFN-γ antibody
For IFN-γ neutralization, anti-mouse IFN-γ antibody (BioXCell; clone XMG1.2) or a rat IgG1 isotype control (BioXCell; clone HRPN) was administered to mice by intraperitoneal injection at Day -1 (0.5 mg) and Day 0 (0.5 mg) relative to MA10 inoculation.

### Harvesting bronchoalveolar lavage
Bronchoalveolar lavage (BAL) was harvested as described[82]. Briefly, after euthanizing the mouse, the animal was placed on its back on a surgical plate. After making an incision in the neck skin near the trachea using a scalpel, the trachea surrounded by sternohyoid muscle was exposed. After placing a cotton thread under the trachea using pincers, the middle of the exposed trachea between two cartilage rings was carefully punctured with a 26 G needle. Then, the catheter of about 0.5 cm was inserted into the trachea and stabilized by tying the trachea around the catheter using the cotton thread placed. A 1 mL syringe loaded with 1 mL of sterile balanced salt solution with 100 μM EDTA was connected to the catheter and the salt/EDTA solution was gently injected into the catheter. The solution was then gently aspirated while massaging the thorax of the mouse. The syringe was removed from the needle and the recovered lavage fluid was transferred into a 15 mL tube placed on ice.

### Cytokine and chemokine measurements
Sera, as well as supernatants were collected by centrifugation of nasal turbinates (NT), bronchioalveolar lavage (BAL), and lung homogenates

(LH) of mice treated with 7DW8-5 one day before. Sera and supernatants were also collected from mice treated with saline one day before. The sera and supernatants were analyzed for cytokines and chemokines by Eve Technologies Corporation (Calgary, AB, Canada) using their Mouse Cytokine /Chemokine 31-Plex Array (MD31).

### Harvesting lung mononuclear cells
Lung mononuclear cells (MNCs) were harvested as described[83]. As shown in Supplementary Fig. 5a, lungs were first perfused with PBS containing collagenase IV and DNase1. Single cell suspensions were prepared by mechanical dissociation of lung tissue through a 70-μm nylon mesh, followed by being layered on Ficoll-Paque (Sigma) and centrifuged at room temperature for 20 min at 900 g. After collecting lung MNC from the gradient interphase, the cells were incubated with anti-mouse Fc receptor antibody (BioLegend, Cat # 101320) to block nonspecific antibody binding. Each sample was then stained with phycoerythrin-labeled anti-mouse CD45 antibody (BioLegend, Cat # 103106) for 30 min. Just before sorting cells, DAPI (×10,000 dilution) was added. Viable cells were sorted by FACS Aria II by gating DAPI negative CD45 positive cells, followed by analyzing using FlowJo v10.8.1 software, as shown in Supplementary Fig. 5b.

### Single cell 5′ transcript and TCRαβ enriched library preparation on the 10x Chromium and sequencing
The scRNA-Seq and scTCR-Seq libraries were prepared using the 10x Chromium single-cell 5′ reagent kits v2 (Dual Index), per manufacturer's instructions. In brief, after cell sorting, single-cell suspensions were loaded into the Chromium controller to make nanoliter-scale droplets with uniquely barcoded 5′ gel beads called GEMs (gel bead-in emulsions). After GEM-RT, GEMs were cleaned up by Dyna beads MyOne Silane beads (Thermo Fisher Scientific, 37002D). According to the manufacturer's instructions, the cDNA was amplified and size-selected by SPRI-beads (Beckman Coulter, B23317). Finally, the 5′ transcript and TCR-seq libraries were pooled and sequenced with the Illumina NovaSeq 6000 system.

### Data processing of scRNA-seq and TCR-seq
FASTQ files were processed by Cell Ranger v.6.1.2 (10x Genomics) software using the GRCm38 (GENCODE vM23/Ensembl 98) genome as a reference. The data analyzed in R using the Seurat v.4.1.0 package were merged and integrated using an anchor-based single-cell data integration method in Seurat[84]. Cells with mitochondrial RNA content greater than 5% were excluded. Variable feature genes were set at 2000 genes. Once the data sets were integrated, the data were input into a principal component analysis (PCA) based on variable genes. The same principal components were used to generate the Uniform Manifold Approximation and Projections (UMAPs). Clusters were identified using shared nearest neighbor–based (SNN-based) clustering. For the clustering analysis, the function RunUMAP, FindClusters, and FindNeighbors in Seurat were used. Single cell TCR sequencing was aligned and quantified using the cellranger-vdj software (v.6.1.2). For the data analysis, the output file filtered_contig_annotations.csv was used.

### Activation of human iNKT cells by 7DW8-5 in vitro
Peripheral blood mononuclear cells (PBMCs) were obtained from 8 healthy donors using Ficoll Paque Plus gradient (Merck Millipore, MA, Cat# GE17-1440-02). PBMCs were added to 96 U-bottom well plates (0.5 × 10^6 cells/well) in duplicates and stimulated with 7DW8-5 at 0.1 and 1 mM final concentration. In parallel, PBMCs were also treated with 18:1 PE at 1 mM final concentration. PBMCs were stimulated for 24 h at 37 °C in 5% CO$_2$ atmosphere. In the final 4 h, Brefeldin A was added at 10 mg/mL and incubated for the 4 additional hours. After that, PBMC supernatants were harvested and stored for analysis of secreted IFN-γ by Luminex (Invitrogen, Waltham, MA) according to the manufacturers' instructions. PBMCs were washed and stained with PE-Cy7

anti-CD3 (clone: SK7), PERCP Cy5.5 anti-TCRVα24 (clone: C15), AF647 anti-CD161 (clone: HP3G10), AF700 anti-CD8 (clone: RPA-T8) and APC-Cy7 anti-CD69 (clone: FN50), for 30 min at room temperature. After staining, cells were permeabilized and stained with anti-human IFN-γ antibody for 40 min at room temperature. After staining, cells were washed with washing solution twice and resuspended in PBS for immediate flow cytometric acquisition. For FACS acquisition, $1 \times 10^5$ lymphocytes were acquired using a BD LSR Fortessa (BD Biosciences, San Jose, CA) Flow Cytometer. Data acquisition was performed using the FACS DIVA software (BD Biosciences, San Jose, CA) and data analysis using FlowJo software (version 10.6.2, BD Biosciences, San Jose, CA).

## In vitro stimulation of human iNKT cell lines by glycolipids

Human iNKT cell lines and HeLa cells transfected with the human CD1d gene, HeLa-hCD1d, were established, as we previously reported[44]. Twenty thousand human iNKT cell lines from 3 different donors were co-cultured with $2 \times 10^4$ HeLa-hCD1d in a well of 96-well round-bottom plate in the presence of 1 μg/mL of 7DW8-5 glycolipid, 18:1 PE or saline (controls). After 24 h incubation, the cultured supernatants were collected and estimated for presence of IFN-γ using ELISA.

## In vitro inhibition of icSARS-CoV-2-mNG-infected Huh7 cells

Supernatants collected from stimulated human iNKT cell lines were serially diluted (2-fold) and incubated on a monolayer of Huh7 cells overnight at 37 C/5%CO2 for 20 h. A serial 5-fold dilution of recombinant human IFN-γ protein was used as positive control (Gibco cat# PHC4031). In addition, supernatants were combined and mixed with anti-human IFN-γ antibody (1 μg/mL) (BioXcell Cat# BE0235) to check for specific and non-specific effects. Following this incubation, the cells were infected with ic-SARS-CoV-2-mNeon isolate[64] and incubated further for 24 h under similar conditions. After 24 h, cells were examined under fluorescent microscope to enumerate GFP⁺ cells and percent inhibition was calculated as the difference in fluorescence in the test wells compared to wells receiving no treatment (virus only) and plotted using GraphPad Prism v9.3.

## Reporting summary

Further information on research design is available in the Nature Portfolio Reporting Summary linked to this article.

## Data availability

Materials used in this study will be made available but may require execution of a materials transfer agreement. All the data are provided in the paper or the Supplementary Information. Source data are provided with this paper. mRNA sequencing data for comparing differential gene expression and RNA-sequencing data associated with single-cell RNA seq experiments have been made available on the European bioinformatics Institute (EBI) server via the ArrayExpress database using accession codes E-MTAB-12345 for mRNA and E-MTAB-11770 for the single cell sequencing. Source data are provided with this paper.

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

## Acknowledgements

This study was supported by funding to D.D.H. from the JPB Foundation, Andrew and Peggy Cherng, Samuel Yin, Carol Ludwig, David and Roger Wu. K18-hACE2 transgenic mouse and hamster experiments were conducted at Washington University with support from NIH: R01 AI157155 (to M.S.D), 75N93019C00051 (to M.S.D.), 75N93021C00016 (to A.C.M.B) and U01 AI151810 (to M.S.D and A.C.M.B).

## Author contributions

M.T., M.S.N., Y.T., Y.H., and D.D.H. conceived the project. M.T., C.C., L.C., and Z.C. performed the mouse experiments. T.L.D., and K.S. performed the hamster experiments. M.S.N., Z.C., T.L.D., K.S., and Y.H. performed the virus titration experiments. K.M. performed the single-cell analyses with 10X Genomics. Y.H. and M.M. performed the histological analysis. A.L.R., and G.M.F. contributed to the flow cytometric assay. M.T., M.S.N, K.M., J.G.A.C.-d.-R, Y.T., A.C.M.B, M.S.D and D.D.H. analyzed the results. M.T. and D.D.H. wrote the manuscript with contributions from each author.

## Competing interests

M.T., M.S.N, Y.H., and D.D.H. are listed as co-inventors on a provisional patent application filed by Columbia University for the treatment for COVID-19 and other viral respiratory infections using 7DW8-5 described in this manuscript. The remaining authors declare no competing interests.
