## [Peer Review File · Nature Communications]

An Immunostimulatory Glycolipid That Blocks SARS-CoV-2, RSV, and Influenza Infections In VivoReviewers' Comments:

Reviewer #1:

Remarks to the Author:

Comments to the author:

The manuscript submitted by Tsuji et al. describes the use of the glycolipid CD1d agonist 7DW8-5 as a tool for rapid anti-viral intervention. The prophylactic and therapeutic application of the compound was tested in mice and hamster following different time schemes in the context of infection with SARS-CoV-2 as well as RSV and influenza infections. These analyses revealed the antiviral potential of 7DW8-5 when given intranasally in a prophylactic setting. Further experiments using knockout mice identified the requirement of CD1d as well as IFN γ for the protective activity.

The described antiviral characteristics of the presented glycolipid antigen 7DW8-5 are very interesting and represent a useful tool for rapid responses in future pandemic settings. The manuscript is well written and the experimental approaches as well as the conclusions are clearly described. I appreciate that the anti-viral potential of the CD1d agonist was tested in different animal models as well as in the context of different viral infections.

Major comments:

- The efficacy of the current vaccines is “undersold”. It should be mentioned that by now vaccines with a good efficacy are on the market and that the strength of the CD1d agonist would lie in the beginning of the pandemic when vaccines and other virus-specific approaches are still under investigation.

- The statement “These cells form a key element of the innate immune response and may play a role not only in cancer and autoimmune diseases, but also in protection against infections” implies that there is only little evidence for the impact of iNKT cells on anti-viral immunity. However, there is a great variety of references showing the importance of iNKT cells in the context of viral infections such as CMV, retroviral infection, RSV and influenza. Most of them are additionally describing the NKT cell-mediated activation of adaptive immune cell populations thereby crucially shaping anti-viral responses. It should be emphasized that the novelty of the reported findings is limited in this context. Here are a few of them: (

<https://doi.org/10.1016/j.cell.2017.11.036>

10.1517/14712590802653601

<https://doi.org/10.1186/s12977-017-0327-8>

<https://doi.org/10.1128/JVI.76.9.4294-4303.2002>

<https://doi.org/10.1016/j.cell.2017.11.036>,

10.1126/sciimmunol.abj8760

<https://doi.org/10.3389/fimmu.2018.00545>

10.3390/vaccines9090949

10.3389/fimmu.2019.01849

10.1038/s41598-019-52666-9

10.1002/eji.201445209.

- The authors introduce their α GalCer analogue. In this context, they should include the description of other analogues that were tested in various contexts and described in various publications.

- In addition to the investigated cytokines, at least IL-4, IL-12 and IL-15 and their interplay with innate immune cells should be mentioned.

- Description of reciprocal activation of DCs and the importance of the DC-derived cytokines secreted due to this activation should be explained.

- It would be very interesting to show more about the different iNKT cell subsets. Which kind iNKT cells are getting activated (iNKT1, iNKT2, iNKT10, iNKT17?). Which of them are important for viral clearance?

- The “mechanism-specific effect of 7DW8-5” is common to other CD1d agonists and was described earlier. This should be mentioned in the introduction or latest in the discussion.

- “The mechanism-specific effect of 7DW8-5” suggests that the observed anti-viral effect is specific for 7DW8-5. However, these effects are rather common for CD1d agonists and were described for other analogues before.
- Fig: 1: The 4th group described to receive 7DW8-5 at two hours post virus challenge is not shown in figure 1b? This is a little bit confusing when this first sub-figure deals with the comparison of the routes and the time points are compared in figure 1c.
- The post-exposure administration of 7DW8-5 showed a partial effect on the tissue viral load. This should be mentioned.
- The aspect of NKT cell anergy should be described in the introduction in the context of other CD1d agonists acting in a similar way.
- Figure 1e: The results shown in figure 1d show that a robust protection was achieved after administration of at least 0.5 ug of the compound while 0.1 ug resulted in suboptimal protection. Why did they choose to use 0.2 ug for this experiment when this dosage was not included in the dose-escalation experiment? Anergy is a dose-dependent process and it would be beneficial to show whether a repeated administration of a higher dose of 7DW8-5 would induce anergy in order to see how robust the result on anergy is. The basis for the argument that 7DW8-5 does not induce anergy is a little weak, especially when for almost all other experiments the shown dose is 2 ug of the compound. This data should be included in the manuscript.
- All figures: For all experiments the number of mice and of independent experiments should be clearly indicated. Some of the results appear to be derived from only five animals and one experiment, which is a very low number for viral challenge experiments and usually higher numbers are required for reliable results.
- Which dosage of 7DW8-5 was used for the histopathologic analyses shown in figure 1g? Is this derived from mice treated with 0.2 ug of the compound? The sample 7DW8-5 (1x) + MA10 shows reduced but obvious staining for SARS-CoV-2 not visible in mice that received repeated administration of 7DW8-5. This should be mentioned and explained/discussed as it seems that here, a suboptimal protection is observed.
- The phrase “...perhaps due to stimulation of a specific immune pathway” is very speculative especially with the knowledge on innate immunity and the fact that it was reported before that the activation of iNKT cells via the DC axis results in the secretion of various cytokines (including IFN γ) thus resulting in a rather unspecific immune activation.
- In figure 1 the authors showed that a dose of 0.1 ug 7DW8-5 conferred suboptimal protection, why was this dose used for this experiment and claimed to confer sufficient protection? Why did the authors not use a higher, reliable dose of the compound in order to elucidate the underlying mechanism?
- First paragraph of “Mechanism of protection”: The relevance of CD1d for the anti-viral effect of 7DW8-5 does not come as a surprise as this was shown before for GalCer and other analogues in various publications. This point should be emphasized latest in the discussion. The same is true for the importance of IFN γ .
- Lines 202-229 is more or less an extensive summary of the findings made by the authors with very little to none reference to any other published manuscript describing important aspects that should be considered when interpreting the presented results. A more extensive discussion in the context of the current literature is required. Here it is very important to consider all relevant publications on GalCer and analogues and proposed mechanisms of action.
- The potential mechanism should be described in more detail and discussed with the current knowledge. It was described before that iNKT cells are activated by lipid antigens bound to CD1d on antigen-presenting cells and that the secretion of IFN γ plays an important role in this mechanism. This and further downstream mechanisms are described and discussed in the literature and should be part of the discussion. E.g., PMID 28860937.
- GalCer is/was already used in clinical trials in humans (see reference above). These were mainly performed in the context of malignancies. However, the mechanism of action will most probably be similar and therefore, these studies should be mentioned here. The authors try to find reasons why they believe that 7DW8-5 will show activity in the human system while this point was already established in clinical trials with GalCer analogs. They should additionally discuss the anti-viral aspect

of their approach and try to discuss this point in the context of current literature.

- Line 251 & 252 indicates that there is only one clinical trial that involved galactosyl ceramide, which is not true.
- In the last paragraph of the discussion, the authors focus the importance of 7DW8-5 on SARS-CoV-2 breakthrough infections. In my opinion, this scenario is a little exaggerated. The authors should additionally focus on e.g., the current vast number of respiratory infections (RSV, influenza, streptococcal infections, ...) that lead to a huge economical loss that could be tackled by the use of an unspecific immune activator such as 7DW8-5.
- It need to be consider that the authors suggest a rather prophylactic but not therapeutic effect of their GalCer analog. A compound that can only be used 2 days prior infection, but not during the course of infection or as a prophylactic vaccine is far beyond a realistic approach against viral infections. Here, a vaccine co-administered with their compound as adjuvant might be much more interesting. A vaccination study should be performed to address this point.

Minor comments:

- Line 43: Spelling error "syncytial"
- Figure 1d: Question marks should be replaced with the correct symbol.
- Figure 1: Which statistical test was used?
- Figure 1 & 2: What does the dotted line in the graphs showing lung viral load indicate?

Reviewer #2:

Remarks to the Author:

In this manuscript the authors study the effect of 7DW8-5, an improved analog of alpha-galactosylceramide, on infection of mice with SARS-CoV-2, RSV and influenza. They report on 7DW8-5's immunostimulatory properties and build on previously published work with the parent molecule, glycolipid alpha-galactosylceramide, and other work on 7DW8-5, applying them here to three different viral infection models. They show that both CD1d receptor and IFN-gamma are necessary for the protective action of 7DW8-5 against SARS-CoV-2 infection in mice and assess the effect of 7DW8-5 in vitro in human cell lines. The authors use intranasal dosing of 7DW8-5, and demonstrate the effect of timing of the dose on in vivo outcomes.

The work is of significant relevance, as broad approaches for emerging/circulating viral infections are lacking, with vaccines, antibodies and small molecule treatments needing specific (and often lengthy) tailoring to pathogens. The authors don't overstate the significance of their observations and acknowledge further work that needs to be done to translate this immunostimulatory therapy.

However, a concern is the novelty of certain findings that is not fully addressed and discussed in the manuscript. First, with respect to mechanism, the requirement of CD1d and IFN-gamma to mediate effects of 7DW8-5 has been reported in different context in previous studies by the authors (cited (20) and uncited: Lee C, Hong SN, Kim YH. A glycolipid adjuvant, 7DW8-5, provides a protective effect against colonic inflammation in mice by the recruitment of CD1d-restricted natural killer T cells. *Intest Res.* 2020;18(4):402-411. doi:10.5217/ir.2019.00132).

Second, there is a lack of a discussion of prior art in antiviral effects of the parent glycolipid alpha-galactosylceramide. There are references to iNKTs and viral infections, but not the glycolipid experiments specifically, and a lack of references to and discussion of manuscripts describing influenza infections in mice and pigs abrogated by alpha-galactosylceramide treatment (with mixed results in pigs, perhaps due to timing of dose or iNKT abundance?) and even effect on TB infection. The manuscript would benefit from such discussion of these previous results, especially the variability of alpha-galactosylceramide treatment in pigs and how the improvements in 7DW8-5's potency (or dosing schedule?) could perhaps overcome the differences in abundance of iNKTs in mice vs humans.

Some manuscripts showing effects of alpha-galactosylceramide on infection include:

Madrid, Darling Melany de C et al. "Comparison of oseltamivir and α -galactosylceramide for reducing disease and transmission in pigs infected with 2009 H1N1 pandemic influenza virus." *Frontiers in veterinary science* vol. 9 999507. 20 Oct. 2022, doi:10.3389/fvets.2022.999507

Artiaga, Bianca L et al. "Rapid control of pandemic H1N1 influenza by targeting NKT-cells." *Scientific reports* vol. 6 37999. 29 Nov. 2016, doi:10.1038/srep37999

Ishikawa, Hiroki et al. "IFN- γ production downstream of NKT cell activation in mice infected with influenza virus enhances the cytolytic activities of both NK cells and viral antigen-specific CD8+ T cells." *Virology* vol. 407,2 (2010): 325-32. doi:10.1016/j.virol.2010.08.030

Ho, Ling-Pei et al. "Activation of invariant NKT cells enhances the innate immune response and improves the disease course in influenza A virus infection." *European journal of immunology* vol. 38,7 (2008): 1913-22. doi:10.1002/eji.200738017

Sada-Ovalle I, Sköld M, Tian T, Besra GS, Behar SM. Alpha-galactosylceramide as a therapeutic agent for pulmonary Mycobacterium tuberculosis infection. *Am J Respir Crit Care Med*. 2010;182(6):841-847. doi:10.1164/rccm.200912-1921OC

There are also a couple minor edits to be addressed:

1. Graph in Fig 1d lost its formatting, with "ug/mL", appearing as "?g/mL"
2. What is the significance of myeloid cell enrichment in 7DW8-5 treated mice (gray points in Fig 3c)?
3. Dashed lines in graphs denoting 0% body weight change are variably used (not present in all graphs). Do the dashed lines in TCID50/g graphs indicate limit of detection?
4. What dose of lipid was given in Fig 4b-d
5. Line 421, is " meant to be there at the end of the line?
6. Extended Fig 6d: how much 7DW8-5 and 18.1 PE was used – assuming 1 ug based on materials, but it might be easier to mention in figure legend. Also, supernatants were serially diluted, but only 1:8 and 1:16 results are shown. Were there negative effects with more concentrated supernatants? At what point did the protection of 7DW8-5 was diluted?

Reviewer #3:

Remarks to the Author:

Tsuji et al. present a very interesting paper showing that a glycolipid that stimulates interferon production can block the replication of SARS-CoV-2, RSV, and influenza infections in vivo. This is potentially quite significant as these are the three major viral respiratory infections that one worries about in the clinic and in terms of epidemics and pandemics. A weakness is that the lipid only works for prophylaxis but even as prophylaxis it could be very useful in view of its broad activity. One major drawback in the paper, which is otherwise very nicely written, is that there is no attention paid to other groups who have used immunostimulatory agents to treat respiratory tract infections. As an example, I would refer you to the works by the laboratory of Scott Evans at MD Anderson. One other consideration concerning how the paper is written: the figures tend to be extremely dense. I suspect that this is due to perhaps a previous submission to a different Nature journal, but Nature Communications allows for more figures, so perhaps the authors could make the figures a little less dense and hence easier to understand. Other concerns are as follows:

1. The authors note that CD1d is present on antigen-presenting cells such as dendritic cells, but they do not note it this is also reportedly present on B cells.
2. As noted by the authors, 7DW8-5 does not work as well against the Delta variant of SARS-CoV-2. This is a little bit surprising so it would be good to have the authors explain a bit more about why they think this is the case.

3. It is quite confusing trying to understand where IFN-gamma is being produced. The authors show that when they look at all compartments IFN-gamma is increased by 7DW8-5, although it is not clear to me how they put the data all together from all compartments. This issue is further confused by the fact that IFN-gamma induction does not appear to be measurable in the supernatants of nasal turbinate and lung homogenates. What do the authors think is going on? Is everything in the BAL fluid? Later in the paper, the authors clearly do demonstrate that IFN-gamma is necessary for the antiviral effect, however.
4. On line 195, I do not understand what "ic" refers to.
5. The authors refer to "MA10" and ultimately it became clear to this reader that this refers to the carrier for the drug, but I believe that this needs to be made clear earlier on unless I missed the explanation.
6. Extended Data Figure 1 is hard to understand. How is this work actually done? It should be explained in the figure legend even if it is noted elsewhere in the paper.
7. Extended Data Figure 6 is difficult to interpret. On one hand, it is quite important to show that this compound will work in humans, but the experiment itself is quite artificial. There is also not that robust of an induction of IFN-gamma, perhaps due to the absence of APCs in this assay.

Reviewer #1 (Remarks to the Author):

Comments to the author:

The manuscript submitted by Tsuji et al. describes the use of the glycolipid CD1d agonist 7DW8-5 as a tool for rapid anti-viral intervention. The prophylactic and therapeutic application of the compound was tested in mice and hamster following different time schemes in the context of infection with SARS-CoV-2 as well as RSV and influenza infections. These analyses revealed the antiviral potential of 7DW8-5 when given intranasally in a prophylactic setting. Further experiments using knockout mice identified the requirement of CD1d as well as IFN γ for the protective activity.

The described antiviral characteristics of the presented glycolipid antigen 7DW8-5 are very interesting and represent a useful tool for rapid responses in future pandemic settings. The manuscript is well written and the experimental approaches as well as the conclusions are clearly described. I appreciate that the anti-viral potential of the CD1d agonist was tested in different animal models as well as in the context of different viral infections.

-We thank the reviewer for their time, positive feedback, and outlook of our work. Please find the responses to their comments below addressed in blue.

Major comments:

- The efficacy of the current vaccines is “undersold”. It should be mentioned that by now vaccines with a good efficacy are on the market and that the strength of the CD1d agonist would lie in the beginning of the pandemic when vaccines and other virus-specific approaches are still under investigation.

- As suggested by this reviewer, we have included a sentence describing that “the strength of the CD1d agonist would lie in the rapid responses to unknown emerging viral infections when vaccines and other virus-specific approaches are still under investigation on page 13, lines 305-307 of the revised manuscript.

- The statement “These cells form a key element of the innate immune response and may play a role not only in cancer and autoimmune diseases, but also in protection against infections” implies that there is only little evidence for the impact of iNKT cells on anti-viral immunity. However, there is a great variety of references showing the importance of iNKT cells in the context of viral infections such as CMV, retroviral infection, RSV and influenza. Most of them are additionally describing the NKT cell-mediated activation of adaptive immune cell populations thereby crucially shaping anti-viral responses. It should be emphasized that the novelty of the reported findings is limited in this context. Here are a few of them:
<https://doi.org/10.1016/j.cell.2017.11.036> Gaya et al. Initiation of Antiviral B Cell Immunity Relies on Innate Signals from Spatially Positioned NKT Cells Cell 2018:
<https://doi.org/10.1517/14712590802653601> Tessmer et al. NKT cell immune responses to viral infection. Expert Opinion on Ther Targets 2009. <https://doi.org/10.1186/s12977-017-0327-8> Littwitz-Salomon et al. NKT cells contribute to the control of acute retroviral infection. Retrovirology 2017. <https://doi.org/10.1128/JVI.76.9.4294-4303.2002> Johnson et al. NKT cells contribute to expansion of CD8+ T cells against RSV. JV 2002. <https://doi.org/10.1126/sciimmunol.abj8760> Cui et al. A circulating subset of iNKT cells mediates antitumor and antiviral immunity Sci Immunol 2022. <https://doi.org/10.3390/vaccines9090949>. <https://doi.org/10.3389/fimmu.2019.01849> Trittel et al. Invariant NKT Cell-Mediated Modulation of ILC1s as a Tool for Mucosal Immune Intervention. Front Immunol 2019. <https://doi.org/10.1002/eji.201445209>. Riese et al. Activated NKT cells imprint NK-cell antiviral activity. Eur J Immunol. 2015.

- Our statement does not intend to discount any prior evidence of iNKT cells presented in context of other viruses. Therefore, we have now included a sentence describing that “there have been a number of studies showing the importance of iNKT cells against viral infections, such as CMV,

retroviral infection, RSV and influenza [citing refs. 22-26], as well as the role of NKT cells in the context of adaptive anti-viral immune response. [citing refs. 27-30] on pages 3-4, lines 67-69 of the revised manuscript.

- The authors introduce their α GalCer analogue. In this context, they should include the description of other analogues that were tested in various contexts and described in various publications.

- We have added a sentence “there have been more than a dozen α -GalCer analogues being described to date [citing refs. 14, 32-43], all of which are able to stimulate iNKT cells in the context of CD1d molecules, resulting in exerting activities against various infections, cancers and auto-immune diseases primarily in a mouse model.” in the introduction paragraph (page 4; lines 70-73).

- In addition to the investigated cytokines, at least IL-4, IL-12 and IL-15 and their interplay with innate immune cells should be mentioned.

- We have included data from our single cell analyses to demonstrate the effect of these cytokines as shown in the figures below.

- Description of reciprocal activation of DCs and the importance of the DC-derived cytokines secreted due to this activation should be explained.

- From our UMAP data (**Fig. 3c**), in which DC subsets were categorized, we have now determined that 30% of mature DCs (mDCs) was found to express IL-15 among lung MNCs 3 days after intranasal administration (2 μ g) of 7DW8-5 (**Figure E** below), whereas mDCs constitutively express IL-12 (data

Figure E: Percentage of DC subsets expressing IL-15 determined from the UMAP of 7DW8-5, comparing to UMAP of saline.

not shown). Plasmacytoid DCs (pDCs) and conventional DCs (cDCs) express minimal amount of IL-15 and IL-12 (data not shown) before and after 7DW8-5 treatment. We present this data in **Supplementary Fig. 6b** of the revised draft and incorporate the results on pages 8-9, lines 192-197 of the revised manuscript.

- It would be very interesting to show more about the different iNKT cell subsets. Which kind iNKT cells are getting activated (iNKT1, iNKT2, iNKT10, iNKT17?). Which of them are important for viral clearance?

- Our data from our single cell analyses experiment was re-analyzed to categorize iNKT cell subsets that express various cytokines in the UMAP of 7DW8-5, comparing to the UMAP of saline. The results presented in **Figure D** above and shown in **Supplementary Fig. 6a** of the revised draft show that a majority of iNKT cell subset, which had expanded among lung MNCs 3 days after intranasal administration (2 μ g) of 7DW8-5, was iNKT1. Therefore, an iNKT1 subset, which secretes IFN- γ ,

appears to predominantly contribute to viral clearance. We have added the results of our analyses into the text on page 8, lines 187-192 of the revised manuscript.

- The “mechanism-specific effect of 7DW8-5” is common to other CD1d agonists and was described earlier. This should be mentioned in the introduction or latest in the discussion.

- We have now mentioned the overlap of mechanism between CD1d agonists in the Introduction (page 4, lines 70-75 of the revised manuscript).

- “The mechanism-specific effect of 7DW8-5” suggests that the observed anti-viral effect is specific for 7DW8-5. However, these effects are rather common for CD1d agonists and were described for other analogues before.

- Our definition of the “mechanism-specific effect” implied that 7DW8-5 exerted its antiviral activity by using CD1d-iNKT cell axis specifically. We now changed the statement to make it more appealing to the reviewer as follows: “CD1d-iNKT cell-dependent effect of 7DW8-5” on page 4, line 80 of the revised manuscript.

- Fig: 1: The 4th group described to receive 7DW8-5 at two hours post virus challenge is not shown in figure 1b? This is a little bit confusing when this first sub-figure deals with the comparison of the routes and the time points are compared in figure 1c.

- The experiments in **Fig. 1b and 1c** were performed separately. To compare the effects of intranasal dosing to intraperitoneal dosing, we administered 7DW8-5 2 days before the virus challenge to obtain the results in **Fig. 1b**. We then measured the duration of the prophylaxis response using our preferred route of intranasal administration of 7DW8-5 in **Fig. 1c**, comparing prophylactic and treatment responses in the mice.

- The post-exposure administration of 7DW8-5 showed a partial effect on the tissue viral load. This should be mentioned.

- We have now mentioned the partial effect of 7DW8-5 post-exposure on page 5, lines 97-98 of the revised manuscript.

- The aspect of NKT cell anergy should be described in the introduction in the context of other CD1d agonists acting in a similar way.

- We have now described NKT cell anergy on page 5, lines 108-110 of the revised manuscript.

- Figure 1e: The results shown in figure 1d show that a robust protection was achieved after administration of at least 0.5 ug of the compound while 0.1 ug resulted in suboptimal protection. Why did they choose to use 0.2 ug for this experiment when this dosage was not included in the dose-escalation experiment? Anergy is a dose-dependent process, and it would be beneficial to show whether a repeated administration of a higher dose of 7DW8-5 would induce anergy in order to see how robust the result on anergy is. The basis for the argument that 7DW8-5 does not induce anergy is a little weak, especially when for almost all other experiments the shown dose is 2 ug of the compound. This data should be included in the manuscript.

- We used 0.2 µg because this dose is two-fold higher than a dose that protects partially (0.1 µg). To alleviate the reviewer’s concern, we have now conducted a repeated dosing of 2µg of 7DW8-5 compared to 0.2 µg head-to-head. As shown in **Figure C** of this document and included in the **Supplementary Fig. 1** of the revised manuscript, there was a slight decrease of antiviral activity, suggesting that there was a partial NKT cell anergy upon a repeat dose of a large dose of the glycolipid.

- • All figures: For all experiments the number of mice and of independent experiments should be clearly indicated. Some of the results appear to be derived from only five animals and one experiment, which is a very low number for viral challenge experiments and usually higher numbers are required for reliable results.

- We wish to note that most, if not all, of the experiments we performed in this study represent one of two similar experiments, as described in the figure legends for **Figs. 1, 2 and 4** of the manuscript.
- Which dosage of 7DW8-5 was used for the histopathologic analyses shown in figure 1g? Is this derived from mice treated with 0.2 ug of the compound? The sample 7DW8-5 (1x) + MA10 shows reduced but obvious staining for SARS-CoV-2 not visible in mice that received repeated administration of 7DW8-5. This should be mentioned and explained/discussed as it seems that here, a suboptimal protection is observed.
- Yes, we administered 0.2 µg of 7DW8-5 for lung histological analyses. We have now mentioned that a single dose of 0.2 µg of 7DW8-5 conferred a partial protection, compared to its repeated dosing, as demonstrated by the number of virus-infected cells in the lung by immunohistochemistry on page 6, lines 126-128 of the revised manuscript.
- The phrase "...perhaps due to stimulation of a specific immune pathway" is very speculative especially with the knowledge on innate immunity and the fact that it was reported before that the activation of iNKT cells via the DC axis results in the secretion of various cytokines (including IFN γ) thus resulting in a rather unspecific immune activation.
- We now changed the phrase to "... perhaps due to activation of immune cascade, followed by a reciprocal stimulation of iNKT cell and CD1d-expressing DC activation" on page 7, lines 163-164 of the revised manuscript.
- In figure 1 the authors showed that a dose of 0.1ug 7DW8-5 conferred suboptimal protection, why was this dose used for this experiment and claimed to confer sufficient protection? Why did the authors not use a higher, reliable dose of the compound in order to elucidate the underlying mechanism?
- Figure 1 was indeed presented to show a comprehensive view of preclinical studies for 7DW8-5 to address its features essential for development of the molecule as an antiviral. Hence, the titrations including those of doses at 0.1ug are presented. But for the rest of the figures, we have indeed administered 2µg of 7DW8-5.
- First paragraph of "Mechanism of protection": The relevance of CD1d for the anti-viral effect of 7DW8-5 does not come as a surprise as this was shown before for GalCer and other analogues in various publications. This point should be emphasized latest in the discussion. The same is true for the importance of IFN γ .
- We have now emphasized this point by including several pertinent publications on pages 3-4, lines 67-73 and lines 80-82.
- Lines 202-229 is more or less an extensive summary of the findings made by the authors with very little to none reference to any other published manuscript describing important aspects that should be considered when interpreting the presented results. A more extensive discussion in the context of the current literature is required. Here it is very important to consider all relevant publications on GalCer and analogues and proposed mechanisms of action.
- Similar to the responses above, we have now included in the introduction in the context of all relevant publications on α -GalCer and analogues and proposed mechanisms of action.
- The potential mechanism should be described in more detail and discussed with the current knowledge. It was described before that iNKT cells are activated by lipid antigens bound to CD1d on antigen-presenting cells and that the secretion of IFN γ plays an important role in this mechanism. This and further downstream mechanisms are described and discussed in the literature and should be part of the discussion. E.g., PMID 28860937.
- We have now included a sentence " α -GalCer is a potent activator of iNKT cells, inducing the production of large amounts of IFN- γ , which helps activate both CD8+ T cells and antigen-presenting

cells (APCs), such as DCs and macrophages [ref. 46] on page 11, lines 250-252 of the revised manuscript.

- GalCer is/was already used in clinical trials in humans (see reference above). These were mainly performed in the context of malignancies. However, the mechanism of action will most probably be similar and therefore, these studies should be mentioned here. The authors try to find reasons why they believe that 7DW8-5 will show activity in the human system while this point was already established in clinical trials with GalCer analogs. They should additionally discuss the anti-viral aspect of their approach and try to discuss this point in the context of current literature.
- We have originally screened and identified 7DW8-5 as a lead compound based on its ability to bind human CD1d and stimulate human iNKT cells *in vitro*. In fact, 7DW8-5 has been shown to have 80-times higher binding affinity to human CD1d than α -GalCer and exert a 140-fold higher dose sparing effect against human iNKT cells than α -GalCer, as shown in our previous publication [ref. 44]. These facts underscore a likelihood that 7DW8-5 shows activity in a human system. We have now emphasized this issue in the discussion (page 12, lines 278-279) of the revised manuscript.
- Line 251 & 252 indicates that there is only one clinical trial that involved galactosyl ceramide, which is not true.
- We have now included more publications related to human trials using α -GalCer. Particularly, we selected the trials in which α -GalCer was directly administered to cancer patients [refs. 70, 71] or hepatitis patients [ref. 72, 73] on page 13, lines 289-291 of the revised manuscript.
- In the last paragraph of the discussion, the authors focus the importance of 7DW8-5 on SARS-CoV-2 breakthrough infections. In my opinion, this scenario is a little exaggerated. The authors should additionally focus on e.g., the current vast number of respiratory infections (RSV, influenza, streptococcal infections, ...) that lead to a huge economical loss that could be tackled by the use of an unspecific immune activator such as 7DW8-5.
- We have now mentioned the potential economic value of 7DW8-5-based prevention against a wide range of respiratory infections (page 13, lines 303- 304 of the revised manuscript).
- It need to be consider that the authors suggest a rather prophylactic but not therapeutic effect of their GalCer analog. A compound that can only be used 2 days prior infection, but not during the course of infection or as a prophylactic vaccine is far beyond a realistic approach against viral infections. Here, a vaccine co-administered with their compound as adjuvant might be much more interesting. A vaccination study should be performed to address this point.
- Indeed, we are considering the use of 7DW8-5 as a vaccine adjuvant. However, this is outside the scope of this manuscript. Nevertheless, we have now briefly discussed about the possibility of using 7DW8-5 as a vaccine adjuvant (page 13-14, lines 306-307 of the revised manuscript).

Minor comments:

- Line 43: Spelling error “syncytial” Fixed the spelling
- Figure 1d: Question marks should be replaced with the correct symbol. Fixed figures to replace the question marks with symbol “ μ ” for $\mu\text{g/mL}$.
- Figure 1: Which statistical test was used? As indicated in the figure legend, non-parametric statistical analysis was done for experiments shown in (b) through (f), using Mann–Whitney U test, and the results represent one of two similar experiments
- Figure 1 & 2: What does the dotted line in the graphs showing lung viral load indicate? The dotted lines in the viral load graph indicates “limit of quantitation” of the viral load and is now mentioned in the figure legend.

Reviewer #2:

In this manuscript the authors study the effect of 7DW8-5, an improved analog of alpha-galactosylceramide, on infection of mice with SARS-CoV-2, RSV and influenza. They report on 7DW8-5's immunostimulatory properties and build on previously published work with the parent molecule, glycolipid alpha-galactosylceramide, and other work on 7DW8-5, applying them here to three different viral infection models. They show that both CD1d receptor and IFN-gamma are necessary for the protective action of 7DW8-5 against SARS-CoV-2 infection in mice and assess the effect of 7DW8-5 in vitro in human cell lines. The authors use intranasal dosing of 7DW8-5 and demonstrate the effect of timing of the dose on in vivo outcomes.

The work is of significant relevance, as broad approaches for emerging/circulating viral infections are lacking, with vaccines, antibodies and small molecule treatments needing specific (and often lengthy) tailoring to pathogens. The authors don't overstate the significance of their observations and acknowledge further work that needs to be done to translate this immunostimulatory therapy.

- We thank the reviewer for their time, positive feedback and appreciation of the work and have responded to their comments below in blue.

However, a concern is the novelty of certain findings that is not fully addressed and discussed in the manuscript. First, with respect to mechanism, the requirement of CD1d and IFN-gamma to mediate effects of 7DW8-5 has been reported in different context in previous studies by the authors [cited (20) and uncited: Lee C, Hong SN, Kim YH. A glycolipid adjuvant, 7DW8-5, provides a protective effect against colonic inflammation in mice by the recruitment of CD1d-restricted natural killer T cells. *Intest Res.* 2020;18(4):402-411].

- We have now added Lee C et al. [ref. 50] on page 4, line 81 of the revised manuscript.

Second, there is a lack of a discussion of prior art in antiviral effects of the parent glycolipid alpha-galactosylceramide. There are references to iNKTs and viral infections, but not the glycolipid experiments specifically, and a lack of references to and discussion of manuscripts describing influenza infections in mice and pigs abrogated by alpha-galactosylceramide treatment (with mixed results in pigs, perhaps due to timing of dose or iNKT abundance?) and even effect on TB infection. The manuscript would benefit from such discussion of these previous results, especially the variability of alpha-galactosylceramide treatment in pigs and how the improvements in 7DW8-5's potency (or dosing schedule?) could perhaps overcome the differences in abundance of iNKTs in mice vs humans.

Some manuscripts showing effects of alpha-galactosylceramide on infection include:

Madrid, Darling Melany de C et al. "Comparison of oseltamivir and α -galactosylceramide for reducing disease and transmission in pigs infected with 2009 H1N1 pandemic influenza virus." *Frontiers in veterinary science* vol. 9 999507. 20 Oct. 2022, doi:10.3389/fvets.2022.999507

Artiaga, Bianca L et al. "Rapid control of pandemic H1N1 influenza by targeting NKT-cells." *Scientific reports* vol. 6 37999. 29 Nov. 2016, doi:10.1038/srep37999

Ishikawa, Hiroki et al. "IFN- γ production downstream of NKT cell activation in mice infected with influenza virus enhances the cytolytic activities of both NK cells and viral antigen-specific CD8+ T cells." *Virology* vol. 407,2 (2010): 325-32. doi:10.1016/j.virol.2010.08.030

Ho, Ling-Pei et al. "Activation of invariant NKT cells enhances the innate immune response and improves the disease course in influenza A virus infection." *European journal of immunology* vol. 38,7 (2008): 1913-22. doi:10.1002/eji.200738017

Sada-Ovalle I, Sköld M, Tian T, Besra GS, Behar SM. Alpha-galactosylceramide as a therapeutic agent for pulmonary Mycobacterium tuberculosis infection. *Am J Respir Crit Care Med.* 2010;182(6):841-847. doi:10.1164/rccm.200912-1921OC

We are very appreciative of the thoughtful comments/suggestions by the reviewer. We are fully aware of the provided references, for example, the parental iNKT cell-stimulating glycolipid, α -GalCer, has been shown to exert anti-microbial activities against influenza virus [Ishikawa et al. *Virology* 2010; Ho et al. *Eur J Immunol* 2008] and TB [Sada-Ovalle et al. *Am J Respir Crit Care Med* 2010]. However, due to

the space limitations in our initial submission, we did not discuss these references in detail. We now do so on page 4, lines 70-73 of the revised draft.

As for the pig model, John Driver's group has previously conducted a very thorough study and published a paper, in which they demonstrated that when α -GalCer, an iNKT cell agonist, not only reduced 2009 H1N1 pandemic influenza virus titers in the upper- and lower-respiratory tract, but also prevented pigs infected with from causing lung inflammation and pathology [Artiaga et al. Sci Rep 2016]. However, the same authors have recently shown that α -GalCer, the same iNKT cell ligand, had no impact (virus replication, lung disease, or virus transmission) on pigs infected with the 2009 H1N1 pandemic influenza virus [Madrid et al. Frontiers in Vet Science 2022]. The authors explain that this discrepancy suggests that the outcome of α -GalCer therapy in swine is unpredictable and may depend on a variety of host factors, such as circulating iNKT-cell concentration, which is highly variable among individual pigs or genetic and environmental factors. Since 7DW8-5 was originally screened and identified based on its potent stimulatory activity against human iNKT cells than α -GalCer (100 times more potent dose sparing effect)[Li et al. PNAS 2010; ref. 44], it would be interesting to test the prophylactic effect of 7DW8-5 in pigs. However, we would like to reserve our comments on the effectiveness of 7DW8-5 in pig model till then.

There are also a couple minor edits to be addressed:

1. Graph in Fig 1d lost its formatting, with "ug/mL", appearing as "?g/mL" - We have fixed figure 1d to reflect the "µ" sign correctly in µg/mL.
2. What is the significance of myeloid cell enrichment in 7DW8-5 treated mice (gray points in Fig 3c)?
 - Activated iNKT cells by CD1d agonists have been shown to stimulate CD1d-expressing monocytes to differentiate myeloid antigen-presenting cells (APCs) [Hedge S. et al. J Autoimmun 2012]. Therefore, it is plausible that 7DW8-5 activated iNKT cells stimulate monocytes to differentiate myeloid APCs here.
3. Dashed lines in graphs denoting 0% body weight change are variably used (not present in all graphs). Do the dashed lines in TCID50/g graphs indicate limit of detection? - We have removed the dashed lines from 0% body weight change on all graphs in Figures 1, 2 and 4 and Supplementary Figures 1 and 7. The dashed lines in viral load graphs indicate the limit of quantitation of the assay and all the figure legends are updated to state so.
4. What dose of lipid was given in Fig 4b-d - Two µg was given and is now indicated in legend of Figure 4 of the revised manuscript.
5. Line 421, is " " meant to be there at the end of the line? - Removed the Quotation mark.
6. Extended Fig 6d: how much 7DW8-5 and 18.1 PE was used – assuming 1 ug based on materials, but it might be easier to mention in figure legend. Also, supernatants were serially diluted, but only 1:8 and 1:16 results are shown. Were there negative effects with more concentrated supernatants? At what point did the protection of 7DW8-5 was diluted?
 - The samples were further diluted but the effects of the 7DW8-5 were decreasing and almost similar to that of negative controls at 1:64 dose. We have now added data for additional dilutions (1:32 and 1:64) to the updated Supplementary Figure 8d in the revised manuscript and have indicated the same in the figure legend.

Reviewer #3:

Tsuji et al. present a very interesting paper showing that a glycolipid that stimulates interferon production can block the replication of SARS-CoV-2, RSV, and influenza infections in vivo. This is potentially quite significant as these are the three major viral respiratory infections that one worries about in the clinic and in terms of epidemics and pandemics. A weakness is that the lipid only works for prophylaxis but even as prophylaxis it could be very useful in view of its broad activity.

-We thank the reviewer for their time and the positive feedback and outlook of the work.

One major drawback in the paper, which is otherwise very nicely written, is that there is no attention paid to other groups who have used immunostimulatory agents to treat respiratory tract infections. As an example, I would refer you to the works by the laboratory of Scott Evans at MD Anderson.

-Some groups have used immunostimulatory agents to treat respiratory tract infections. For example, mice treated before or soon after infection with a combination of inhaled Toll-like receptor (TLR) 2/6 and 9 agonists (Pam2-ODN) were shown to be broadly protected against microbial pathogens including respiratory viruses [S Wali & Evans SE et al. AJRCMB, 2020; ref. 9]. This sentence was added on page 3, lines 55-59 of the revised manuscript.

One other consideration concerning how the paper is written: the figures tend to be extremely dense. I suspect that this is due to perhaps a previous submission to a different Nature journal, but Nature Communications allows for more figures, so perhaps the authors could make the figures a little less dense and hence easier to understand.

-The reviewer is correct that the figures were created to be submitted to a different journal initially, but we do believe that each composite figure presents a summary of the aspect covered by the figure. For example: **Fig. 1** presents the pharmacodynamics of the molecule in vivo, **Fig. 2** presents the overview of the breadth, **Fig. 3** and **Fig. 4** delves to detail the role of CD1d and IFN directed responses.

Other concerns are as follows:

1. The authors note that CD1d is present on antigen-presenting cells such as dendritic cells, but they do not note it this is also reportedly present on B cells.

-We have now added that CD1d is present on B cells too in the introduction (page 3, line 66 of the revised manuscript).

2. As noted by the authors, 7DW8-5 does not work as well against the Delta variant of SARS-CoV-2. This is a little bit surprising so it would be good to have the authors explain a bit more about why they think this is the case

-Both the number and place of hACE2 expression in K18-hACE2 Tg mice is shown to be distinctly different compared to endogenous mACE2 expression in wild-type mice. With regards to the Delta variant results, therefore, hACE2 overexpression in K18-hACE2 Tg mice could render viral infection more efficient, making it as such, hard to protect (less effective protection). In addition, K18 promoter leads to an artificial hACE2 expression by all epithelia [Winkler ES et al. Nat Immunol 2020; ref. 57], where iNKT cells may not be able to access to. We now added this on page 7, lines 147-149 of the revised manuscript. Regardless, we would like to emphasize that although protection against the Delta variant in K18-hACE2 Tg mice and hamsters seems to be less robust, we observed a statistically significant degree of protection (from both viral load reduction and body weight loss).

3. It is quite confusing trying to understand where IFN-gamma is being produced. The authors show that when they look at all compartments IFN-gamma is increased by 7DW8-5, although it is not clear to me how they put the data all together from all compartments. This issue is further confused by the fact that IFN-gamma induction does not appear to be measurable in the supernatants of nasal turbinate and lung homogenates. What do the authors think is going on? Is everything in the BAL fluid? Later in the paper, the authors clearly do demonstrate that IFN-gamma is necessary for the antiviral effect, however.

- We apologize for the confusion. In fact, we showed that IFN-g production was significantly upregulated in the nasal turbinates and lung homogenates from our Luminex data shown in **Fig. 3b**. We do realize that this confusion stemmed from the cropping out of the legend key of **Fig. 3b** on the right that showed the color match of each bar to the designated tissue that was processed to obtain the corresponding data on the multiplex. The legend key has now been added back to **Fig. 3b** to fix the issue.

4. On line 195, I do not understand what “ic” refers to.

- “ic” refers to infectious clone. The construct used in the experiment is not a natural viral isolate but an infectious clone that was generated in the citation [Xie et al. Cell host microbe 2020; ref. 65]. The citation is now added next to the name of the clone in that sentence to indicate its source on page 10, line 227 of the revised manuscript. We have also indicated the source of the virus in the Supplementary file section of materials and methods under cells and viruses on page 2, lines 56- 59 of the revised manuscript.

5. The authors refer to “MA10” and ultimately it became clear to this reader that this refers to the carrier for the drug, but I believe that this needs to be made clear earlier on unless I missed the explanation.

- MA10 is the name of the virus isolate that was adapted to be infectious in mice and has been extensively used for studying SARS-CoV-2 in mice (ref. 51 of the revised manuscript). The source of this virus (BEI) is mentioned in the Supplementary file methods section on “cells and viruses” line 54 of the revised manuscript.

6. Extended Data Figure 1 is hard to understand. How is this work actually done? It should be explained in the figure legend even if it is noted elsewhere in the paper.

- This quantitation of the immunohistochemistry data shown in **Fig. 1g** was performed using Image J software from the stained slides for each treatment. Each slide was counted for the number of staining-positive spots in an imaged slide area of 0.8 mm². A table of the total number of the images counted for each treatment (6 images each) has now been added to **Supplementary Fig. 2** in the revised manuscript and is also shown below.

Samples	Spot number of virus-infected cells in lung IHC			
	Left	Right	Mean	SD
Slide 1 (Saline)	0	0	0	0
Slide 2 (Saline)	0	0		
Slide 3 (Saline)	0	0		
Slide 4 (MA10)	334	297	464.5	188.2
Slide 5 (MA10)	665	254		
Slide 6 (MA10)	610	627		
Slide 7 (7DW8-5 x 1)	0	0	0	0
Slide 8 (7DW8-5 x 1)	0	0		
Slide 9 (7DW8-5 x 1)	0	0		
Slide 10 (7DW8-5 x 1 + MA10)	68	23	41.3	22.8
Slide 11 (7DW8-5 x 1 + MA10)	42	26		
Slide 12 (7DW8-5 x 1 + MA10)	70	19		
Slide 13 (7DW8-5 x 5)	0	0	0	0
Slide 14 (7DW8-5 x 5)	0	0		
Slide 15 (7DW8-5 x 5)	0	0		
Slide 16 (7DW8-5 x 5 + MA10)	201	45	70.3	65.6
Slide 17 (7DW8-5 x 5 + MA10)	30	67		
Slide 18 (7DW8-5 x 5 + MA10)	51	28		

7. Extended Data Figure 6 is difficult to interpret. On one hand, it is quite important to show that this compound will work in humans, but the experiment itself is quite artificial. There is also not that robust of an induction of IFN-gamma, perhaps due to the absence of APCs in this assay.

- Supplementary Figure 6 (now **Supplementary Fig. 8**) is generated by creating iNKT cell lines [Li et al. PNAS 2010; ref 44] that have been stimulated with either PE (without any sugar) or 7DW8-5. The supernatant collected from the activation of these clonal population is expected to have measurable amounts of IFN- γ as has been shown in the inset table. However, even with the levels of IFN- γ , as indicated in the ELISA data, the levels of viral inhibition in the Huh7 human cells receiving the supernatant prior to infection are appreciable. We have expanded the data plot to show further dilutions of the dose response levels. We do appreciate that it is an in vitro experimental setting and further studies in humans will be essential to recapitulate the effects observed in vitro and mice.

Reviewers' Comments:

Reviewer #1:

Remarks to the Author:

No further comments to the authors, all questions and comments were answered to my satisfaction.

Reviewer #2:

Remarks to the Author:

This revised version of the manuscript has addressed my main and minor concerns. I appreciate the authors' discussion in the rebuttal regarding the response of the pig model to α -GalCer and changes to the manuscript made based on comments from other reviewers. My recommendation would be for the revised manuscript to be accepted for publication.

Response to reviewer comments for NCOMMS-23-03998-A

Reviewer #1 (Remarks to the Author):

No further comments to the authors, all questions and comments were answered to my satisfaction.

Thank you.

Reviewer #2 (Remarks to the Author):

This revised version of the manuscript has addressed my main and minor concerns. I appreciate the authors' discussion in the rebuttal regarding the response of the pig model to α -GalCer and changes to the manuscript made based on comments from other reviewers. My recommendation would be for the revised manuscript to be accepted for publication.

Thank you.